# Zika virus causes placental pyroptosis and associated adverse fetal outcomes by activating GSDME

Zikai Zhao[1,2,3,4], Qi Li[1,2,3,4], Usama Ashraf[2], Mengjie Yang[1,2,3,4], Wenjing Zhu[1,2,3,4], Jun Gu[1,2,3,4], Zheng Chen[1,2,3,4], Changqin Gu[4], Youhui Si[1,2,3,4], Shengbo Cao[1,2,3,4]*, Jing Ye[1,2,3,4]*

[1]State Key Laboratory of Agricultural Microbiology, Huazhong Agricultural University, Wuhan, China; [2]Key Laboratory of Preventive Veterinary Medicine in Hubei Province, College of Veterinary Medicine, Huazhong Agricultural University, Wuhan, China; [3]The Cooperative Innovation Center for Sustainable Pig Production, Huazhong Agricultural University, Wuhan, China; [4]College of Veterinary Medicine, Huazhong Agricultural University, Wuhan, China

**Abstract** Zika virus (ZIKV) can be transmitted from mother to fetus during pregnancy, causing adverse fetal outcomes. Several studies have indicated that ZIKV can damage the fetal brain directly; however, whether the ZIKV-induced maternal placental injury contributes to adverse fetal outcomes is sparsely defined. Here, we demonstrated that ZIKV causes the pyroptosis of placental cells by activating the executor gasdermin E (GSDME) in vitro and in vivo. Mechanistically, TNF-α release is induced upon the recognition of viral genomic RNA by RIG-I, followed by activation of caspase-8 and caspase-3 to ultimately escalate the GSDME cleavage. Further analyses revealed that the ablation of GSDME or treatment with TNF-α receptor antagonist in ZIKV-infected pregnant mice attenuates placental pyroptosis, which consequently confers protection against adverse fetal outcomes. In conclusion, our study unveils a novel mechanism of ZIKV-induced adverse fetal outcomes via causing placental cell pyroptosis, which provides new clues for developing therapies for ZIKV-associated diseases.

*For correspondence:
sbcao@mail.hzau.edu.cn (SC);
yej@mail.hzau.edu.cn (JY)

Competing interest: The authors declare that no competing interests exist.

## Editor's evaluation

The study of Zika virus-induced cell death is critical for our understanding of viral pathogenesis. The authors have studied the molecular mechanisms by which Zika virus causes cell death.

## Introduction

Zika virus (ZIKV), a mosquito-borne flavivirus, was initially identified in 1947 but received little concern until it posed a serious public health threat in the Pacific from 2007 to 2015 (*Petersen et al., 2016*). ZIKV exhibits obvious tropism and infects several immunologically privileged regions that include male and female reproductive organs, adult and fetal central/peripheral nervous system, urinary tract, and the structural and neural portions of the eye (*Zimmerman et al., 2020*). Clinically, most ZIKV cases are asymptomatic or mild with low-grade fever, rash, arthralgia, myalgia, and conjunctivitis (*Duffy et al., 2009*; *Musso and Gubler, 2016*). Nevertheless, the ZIKV infection of pregnant women still remains a major concern because the placental infection and the vertical transmission of the virus can cause adverse effects on the fetus, such as congenital ZIKV syndrome (CZS) which is characterized by

microcephaly, intrauterine growth restriction, spontaneous abortion, and developmental abnormalities (*Miner et al., 2016*).

Although several studies have explicitly indicated a causal relationship between ZIKV infection and CZS (*Brasil et al., 2016*; *Mlakar et al., 2016*; *Yockey et al., 2016*), the underlying mechanism is not completely elucidated. On one hand, ZIKV can infect the fetus through the transplacental route, which can lead to CZS in all trimesters of pregnancy (*Brasil et al., 2016*; *Hoen et al., 2018*; *Reynolds et al., 2017*). Studies using the brain organoids, neurospheres, and human pluripotent stem cell-derived brain cells have identified that neural stem and progenitor cells can undergo growth and developmental aberrations upon ZIKV infection, thereby resulting in microcephaly (*Tang et al., 2016*; *Cugola et al., 2016*; *Garcez et al., 2016*). On the other hand, the ZIKV infection-caused placental injury also likely contributes to adverse fetal outcomes. Several studies have demonstrated that ZIKV infects placental macrophages and trophoblasts, which leads to placental insufficiency and injury (*Quicke et al., 2016*; *Szaba et al., 2018*; *Brown et al., 2019*). Using the mouse pregnancy models, it has been shown that ZIKV infection disrupts the architecture and function of the placenta by triggering trophoblast apoptosis and vascular endothelial cell damage (*Miner et al., 2016*; *Yockey et al., 2016*; *Ribeiro et al., 2018*; *Yockey et al., 2018*). Considering that placenta is infected prior to the fetus and that placenta is indispensable for the maternal-fetal nutrient exchange, it is reasonable to speculate that the placental damage during ZIKV infection could be more decisive for adverse fetal outcomes.

During ZIKV infection, the placenta can suffer from apoptosis (*Miner et al., 2016*; *Yockey et al., 2018*), which is a prototype of programmed cell death (PCD) and is crucial for reproduction, embryonic development, immunity, and viral infections. Therefore, it is meaningful to explore whether other PCD processes, such as pyroptosis, are involved in the ZIKV-induced placental damage. Pyroptosis is an innate immune response against intracellular pathogens, and is featured by cell swelling and the emergence of large bubble-like structures from the plasma membrane that differentiate it from other PCDs (*Shi et al., 2017*; *Jorgensen and Miao, 2015*). Recent studies have identified gasdermin D (GSDMD) as a key pyroptosis executor, whose N-terminus domain is capable of binding to membrane lipids, phosphoinositides, and cardiolipin in order to perforate the membrane, resulting in the release of cytosolic content (*Ding et al., 2016*; *Shi et al., 2015*; *Kayagaki et al., 2015*). Gasdermin E (GSDME), also known as DFNA5, belongs to the same superfamily as GSDMD, and was originally identified as a gene related to hearing impairment (*Van Laer et al., 1998*). GSDME is considered as a potential tumor suppressor gene in several types of cancers, such as gastric and hepatocellular carcinomas (*Akino et al., 2007*; *Wang et al., 2013*; *Zhang et al., 2020*) Recently, it has also been demonstrated to mediate pyroptosis via its activated N-terminus (GSDME-N) domain that is specifically cleaved by caspase-3 and granzyme B (*Zhang et al., 2020*; *Rogers et al., 2017*). GSDME can switch the caspase-3-mediated apoptosis to pyroptosis or can mediate secondary necrosis after apoptosis (*Wang et al., 2017*), indicating an extensive crosstalk between apoptosis and GSDME-mediated pyroptosis.

In the context of viral infections, pyroptosis is one of the imperative pathogenic mechanisms that occur during flavivirus infections. For instance, dengue virus can induce the NLRP3 inflammasome-dependent pyroptosis in human macrophages via the C-type lectin 5A, which is critical for dengue hemorrhagic fever (*Wu et al., 2013*; *Chen et al., 2008*). Japanese encephalitis virus enhances the pyroptosis of peritoneal macrophages to raise the serum interleukin-1α level, which in turn promotes viral neuroinvasion and blood-brain-barrier disruption (*Wang et al., 2020*). Particularly, a recent study demonstrated the effect of ZIKV in inducing the caspase-1- and GSDMD-mediated pyroptosis of neural progenitor cells that contributes to CZS (*He et al., 2020*). In view of all these studies, it is worth exploring whether pyroptosis participates in the ZIKV-induced placental damage and adverse fetal outcomes.

Here, we report that ZIKV infection activates caspase-3 via the caspase-8-mediated extrinsic apoptotic pathway, which in turn activates GSDME to induce the pyroptosis of placental trophoblasts. This phenomenon was found to associate with the ZIKV genomic RNA, but not with the ZIKV structural or nonstructural (NS) proteins, upon its recognition by the RIG-I sensor. By establishing a mouse model of ZIKV infection, we further showed that the deletion of GSDME leads to the reduction of placental damage and associated adverse fetal outcomes in infected pregnant mice. This study provides novel insights into the mechanism of ZIKV-induced placental damage and CZS; therefore, it may lead to the development of new therapeutic options to avert adverse fetal outcomes and/or miscarriages during ZIKV infection.

# Results

## ZIKV infection induces the GSDME-mediated pyroptosis in JEG-3 cells

Previous studies have shown that ZIKV infection during the first trimester of pregnancy leads to a higher chance of CZS, but the pathogenesis is yet to be further studied (*Musso and Gubler, 2016*; *Jagger et al., 2017*). Since the ZIKV-infected placenta shows obvious apoptosis of trophoblasts (*Miner et al., 2016*), it prompted us to hypothesize whether other routes of cell death are triggered in response to ZIKV infection. To this end, we chose a ZIKV-permissive human choriocarcinoma cell line JEG-3, which represents the features of trophoblasts and is considered a suitable in vitro model to study the first-trimester placental function (*Zachariades et al., 2011*). Upon ZIKV infection, JEG-3 cells showed evident cell swelling with balloon-like structures originating from the plasma membrane that appears to be distinct from the apoptotic blebbing (*Figure 1A*), displaying a characteristic pyroptotic cell morphology. Along with decreased cell viability, the swelling cells were also found to release lactate dehydrogenase (LDH, an indicator of the lytic cell cytotoxicity) in an infection time-dependent manner, which depicts the rupture and leakage of the plasma membrane (*Figure 1B*). We next determined whether the pyroptosis executors, GSDMD and GSDME, were activated during the infection of JEG-3 cells with ZIKV. To this end, the infected cells were subjected to Western blot analysis, and the results revealed that infection led to the release of the N-terminal domain of GSDME (GSDME-N), whereas no cleavage of GSDMD was observed (*Figure 1C*), implying that ZIKV infection may induce pyroptosis through GSDME.

To further confirm the role of GSDME in ZIKV-induced pyroptosis, a GSDME-knockout (KO) JEG-3 cell line was generated by employing the CRISPR/Cas9 system (*Figure 1D*). As expected, deletion of GSDME significantly suppressed the ZIKV-induced cell swelling (*Figure 1E*) as well as LDH release and the decline of cell viability (*Figure 1F*). In addition, to explore whether GSDME-mediated pyroptosis could affect ZIKV replication, the plaque assay was conducted in GSDME-KO and -overexpressed JEG-3 cells, and the results showed that deletion or overexpression of GSDME did not affect ZIKV replication in JEG-3 cells (*Figure 1—figure supplement 1*), suggesting that GSDME-mediated pyroptosis may be primarily a pathogenic process rather than an antiviral mechanism during ZIKV infection. Overall, these findings demonstrate that ZIKV infection induces the pyroptosis of JEG-3 cells in a GSDME-mediated manner.

## Both the cellular GSDME abundance and the susceptibility to ZIKV infection determine the occurrence of pyroptosis

Since there is no previous study reporting that ZIKV can induce GSDME-mediated pyroptosis, we asked whether it is a common cellular process during ZIKV infection. To answer this question, five different human tissue-derived cell lines, including Huh-7 (liver), A549 (lung), HeLa (cervix), HEK-293T (kidney), and SH-SY5Y (neuroblasts) were infected with ZIKV, and their cytopathies were subsequently examined. It was observed that Huh-7 and A549 cells showed distinct pyroptotic cell death, elevated LDH levels, and GSDME cleavage upon ZIKV infection, whereas HeLa, HEK-293T, and SH-SY5Y cells exhibited no obvious cytopathic effects even at 72 hr post-infection (*Figure 2A-C*). It has been well known that the GSDME-mediated pyroptosis is closely related to the GSDME abundance (*Wang et al., 2017*). However, we noticed that infected SH-SY5Y cells did not undergo pyroptosis despite expressing a high level of GSDME, while Huh-7 and A549 cells expressing the relatively lower GSDME presented the different degrees of pyroptosis and GSDME activation upon infection (*Figure 2A-C*, and *Figure 2—figure supplement 1A*), suggesting that other factors are involved in ZIKV-induced pyroptosis.

In the context of infection, all these cells showed marked discrepancies in terms of susceptibility to ZIKV (*Ramos da Silva et al., 2019*; *Chan et al., 2016*). Among them, JEG-3, Huh-7, and A549 cells displayed the highest susceptibility to ZIKV infection, as assessed by the immunofluorescence assay and the plaque assay (*Figure 2—figure supplement 1B and C*). The observed positive correlation between the cellular susceptibility to infection and the increased cytopathic effects indicates that the cellular susceptibility to ZIKV infection is probably another key factor that determines the occurrence of pyroptosis (*Figure 2D*). To verify whether more viral particles could lead to graver pyroptosis, JEG-3 cells were infected with ZIKV at an increasing multiplicity of infection (MOI). At 24 hr post-infection, the ZIKV-induced LDH release (*Figure 2E*) and GSDME activation (*Figure 2F*) were found to be enhanced in a virus dose-dependent manner, which implicit that the productive infection is critical

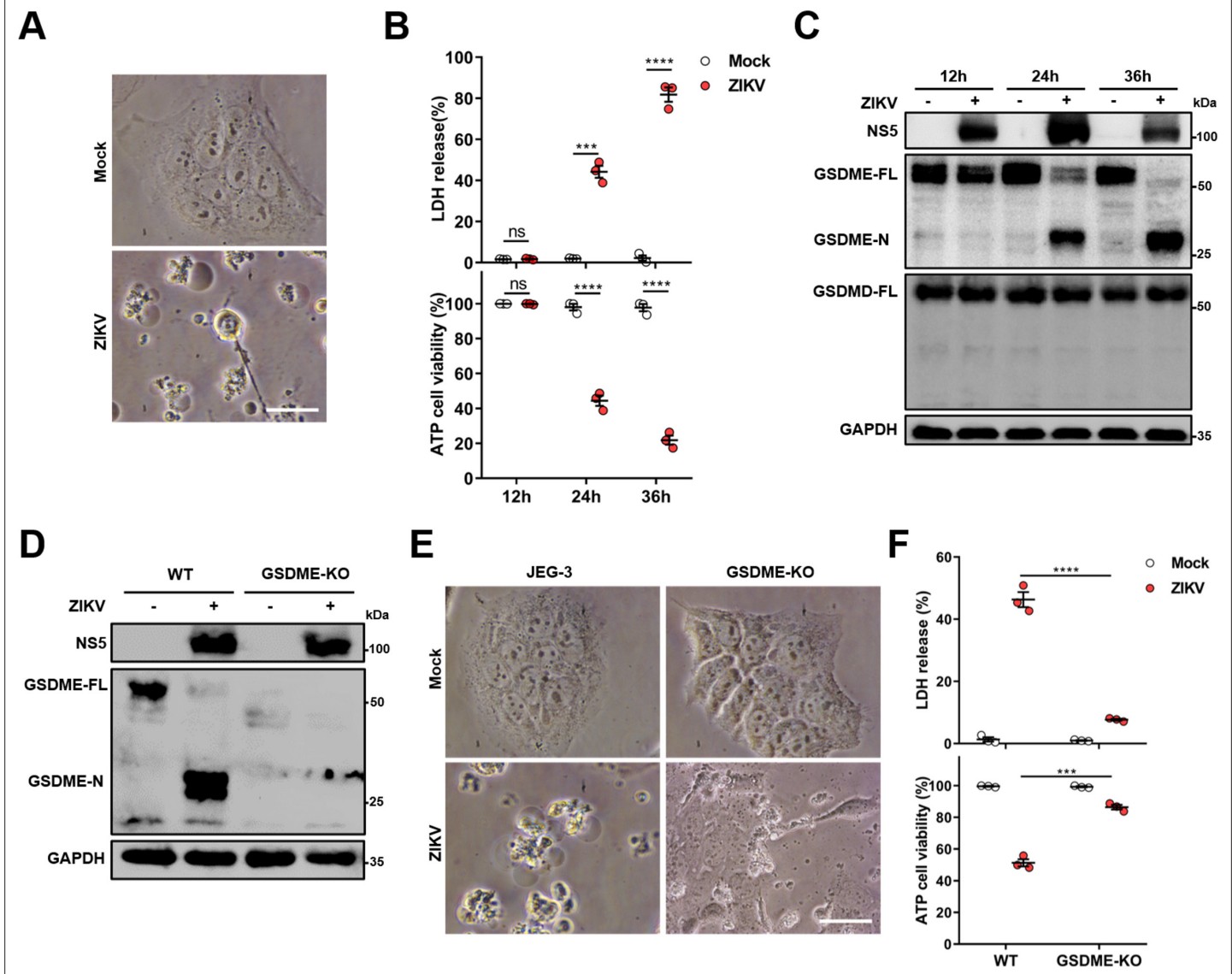

**Figure 1.** Zika virus (ZIKV) infection induces the gasdermin E (GSDME)-mediated pyroptosis in JEG-3 cells. JEG-3 or GSDME-KO JEG-3 cells were infected with ZIKV at a multiplicity of infection (MOI) of 1. At indicated time post-infection, cells were subjected to microscopy, cytotoxicity, and Western blot analyses. (**A**) JEG-3 cells were infected with ZIKV for 24 hr. Representative cell morphology was shown. Scale bar, 50 µm. (**B**) LDH levels in supernatant and cell viability were measured at indicated time post-infection (n=3). (**C**) Immunoblot analyses of GSDME-FL, GSDME-N, and GSDMD-FL in ZIKV-infected JEG-3 cells at indicated time post-infection. (**D–F**) JEG-3 and GSDME-KO JEG-3 cells were infected with ZIKV for 24 hr. Immunoblot analyses of GSDME-FL and GSDME-N by Western blot (**D**). Representative cell morphology was shown. Scale bar, 50 µm (**E**). LDH levels in supernatant and cell viability were measured (n=3) (**F**). LDH release (B, F) is presented as the mean ± SEM of three independent experiments, and the two-tailed unpaired Student's t-test was used to calculate significance. ***, p<0,001; ****, p<0.0001; ns, no significance.

The online version of this article includes the following source data and figure supplement(s) for figure 1:

**Source data 1.** Raw data for *Figure 1*.

**Figure supplement 1.** Gasdermin E (GSDME) does not affect Zika virus (ZIKV) replication in JEG-3 cells.

**Figure supplement 1—source data 1.** Raw data for *Figure 1—figure supplement 1*.

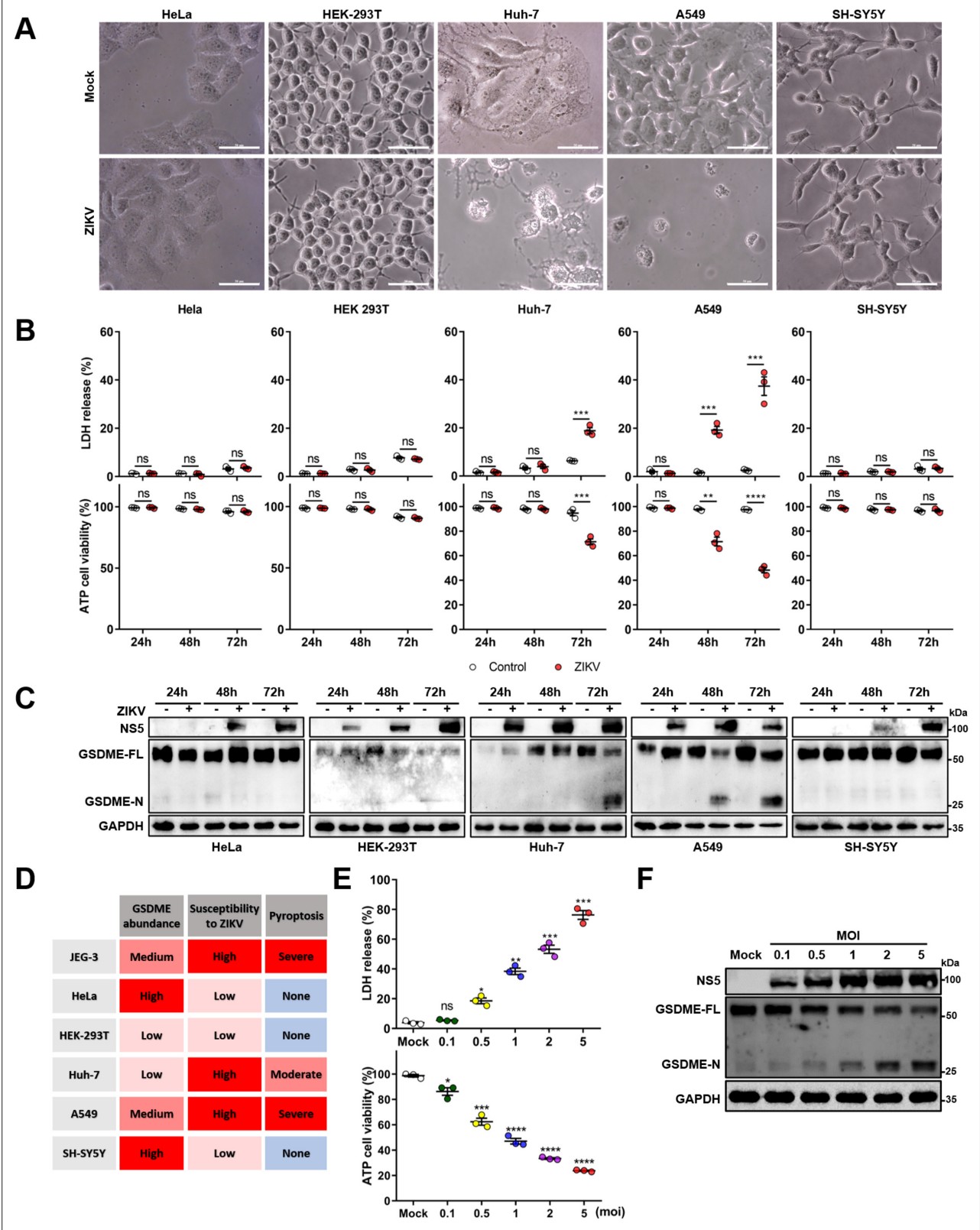

**Figure 2.** Both the cellular gasdermin E (GSDME) abundance and the susceptibility to Zika virus (ZIKV) infection determine the occurrence of pyroptosis. (A–C) Relevant cells were infected with ZIKV at a multiplicity of infection (MOI) of 1. At indicated time post-infection, cells were subjected to microscopy, cytotoxicity, and Western blot analyses. Phase-contrast images of ZIKV-induced pyroptotic cell death in HeLa, HEK 293T, Huh-7, A549, and SH-SY5Y cells at 72 hr post-infection. Scale bar, 50 µm (**A**). Comparison of ATP cell viability, LDH release-based cell death (n=3) (**B**), and GSDME cleavage (**C**) in ZIKV-

*Figure 2 continued on next page*

*Figure 2 continued*

infected relevant cells. (**D**) Table summarizing results shown in (**A–C**), and *Figure 2—figure supplement 1* from ZIKV infection experiments with relevant cell lines. The abundance of GSDME, the replication and infection levels of ZIKV in cells, and the ZIKV-induced LDH release were quantified. The cells showing the highest abundance of GSDME (SHSY-5Y), susceptibility to ZIKV (JEG-3), and degree of pyroptosis (JEG-3) were considered as references. Those cells are defined as high (severe) if their corresponding index is higher than the 75% of reference, medium (moderate) if the index is between 75% and 25%, and low (none) when the index is less than the 25% of the reference. (**E and F**) Analyses of ATP cell viability, LDH release-based cell death (n=3) (**E**), and GSDME cleavage (**F**) in JEG-3 cells at 36 hr post-infection. Unpaired t-test versus mock. LDH release (B, E) is presented as the mean ± SEM of three independent experiments, and the two-tailed unpaired Student's t-test was used to calculate significance. *, $p<0.05$; **, $p<0.01$; ***, $p<0.001$; ****, $p<0.0001$; ns, no significance.

The online version of this article includes the following source data and figure supplement(s) for figure 2:

**Source data 1.** Raw data for *Figure 2*.

**Figure supplement 1.** The gasdermin E (GSDME) abundance and the susceptibility to Zika virus (ZIKV) infection of different cell lines.

**Figure supplement 1—source data 1.** Raw data for *Figure 2—figure supplement 1*.

to its pathogenesis. Collectively, these results indicate that the ZIKV-GSDME-mediated pyroptosis is more likely to occur in those cells that are susceptible to ZIKV infection, and with relatively higher GSDME abundance.

## ZIKV infection activates GSDME via extrinsic apoptotic pathway

It is known that GSDME is cleaved and activated specifically by caspase-3, which consequently results in pyroptosis (*Rogers et al., 2017*; *Wang et al., 2017*). Hence, we explored the activation of caspase-3 upon ZIKV infection. At 24 hr post-infection, caspase-3 underwent activation and obvious cleavage of GSDME was shown in infected JEG-3 cells (*Figure 3—figure supplement 1A*). To verify if other cell death pathways are involved in the ZIKV-induced pyroptosis, ZIKV-infected JEG-3 cells were incubated with caspase-3 inhibitor (Z-DEVD-FMK), caspase-1 inhibitor (VX-765), pan-caspase inhibitor (Z-VAD-FMK), or necrosis inhibitor (GSK872). As shown in *Figure 3A and B*, ZIKV infection caused the induction of pyroptotic morphological features, and the cell death was partially prevented in the presence of caspase-3 or pan-caspase inhibitors. In line with these findings, the Western blot analysis revealed that caspase-3 or pan-caspase inhibitors significantly attenuated the concomitant cleavage of caspase-3 and GSDME (*Figure 3C* and *Figure 3—figure supplement 1B*). In contrast, the treatment of cells with caspase-1 inhibitor and necrosis inhibitor showed no effect on the cell death caused by ZIKV (*Figure 3A-C*), which suggests that necrosis and GSDMD-mediated pyroptosis were not involved in it. These data indicate that ZIKV infection facilitates the caspase-3-dependent cleavage of GSDME to elicit pyroptotic cell death.

Before the caspase-3 was reported as a key regulator of the GSDME-mediated pyroptosis, it was considered an essential effector at the end of apoptotic cascades. During apoptosis, caspase-3 can be activated by intrinsic and/or extrinsic pathways: the former refers to the permeabilization of the mitochondrial membrane and the assembly of apoptosome resulting in the activation of caspase-9, and the latter involves the activation of death receptors and caspase-8 (*Man and Kanneganti, 2016*; *Nagata, 2018*). It has been previously demonstrated that the GSDME-dependent pyroptosis undergoes activation downstream of the intrinsic apoptotic pathway and can potentially be activated downstream of the extrinsic apoptosis pathway (*Ribeiro et al., 2018*; *Yu et al., 2019*; *Lu et al., 2018*). However, the engagement of intrinsic and/or extrinsic pathways to regulate the GSDME-dependent pyroptosis during ZIKV infection remains to be studied. To delineate the molecular mechanism by which cellular pyroptosis was instigated under ZIKV infection, we detected the activation of caspase-8 and caspase-9 in infected JEG-3 cells by the Western blot analysis. As expected, ZIKV infection triggered the activation of both caspase-8 and caspase-9 (*Figure 3—figure supplement 1C*), revealing the fact that various biological processes are triggered during ZIKV infection. Next, the ZIKV-infected cells were incubated with caspase-8 inhibitor (Z-IETD-FMK) or caspase-9 inhibitor (Z-LEHD-FMK), and subsequently were subjected to the analysis of pyroptotic cell parameters. It was found that caspase-8 inhibitor remarkably inhibited the cell membrane rupture, LDH release, and GSDME cleavage in infected cells, while inhibition of caspase-9 only slightly inhibited ZIKV-induced activation of GSDME and did not significantly affect ZIKV-induced lytic cell death (*Figure 3D-F*, and *Figure 3—figure supplement 1D*), suggesting caspase-8 rather than caspase-9 may play a major role in the activation of caspase-3 and GSDME during ZIKV infection in JEG-3 cells.

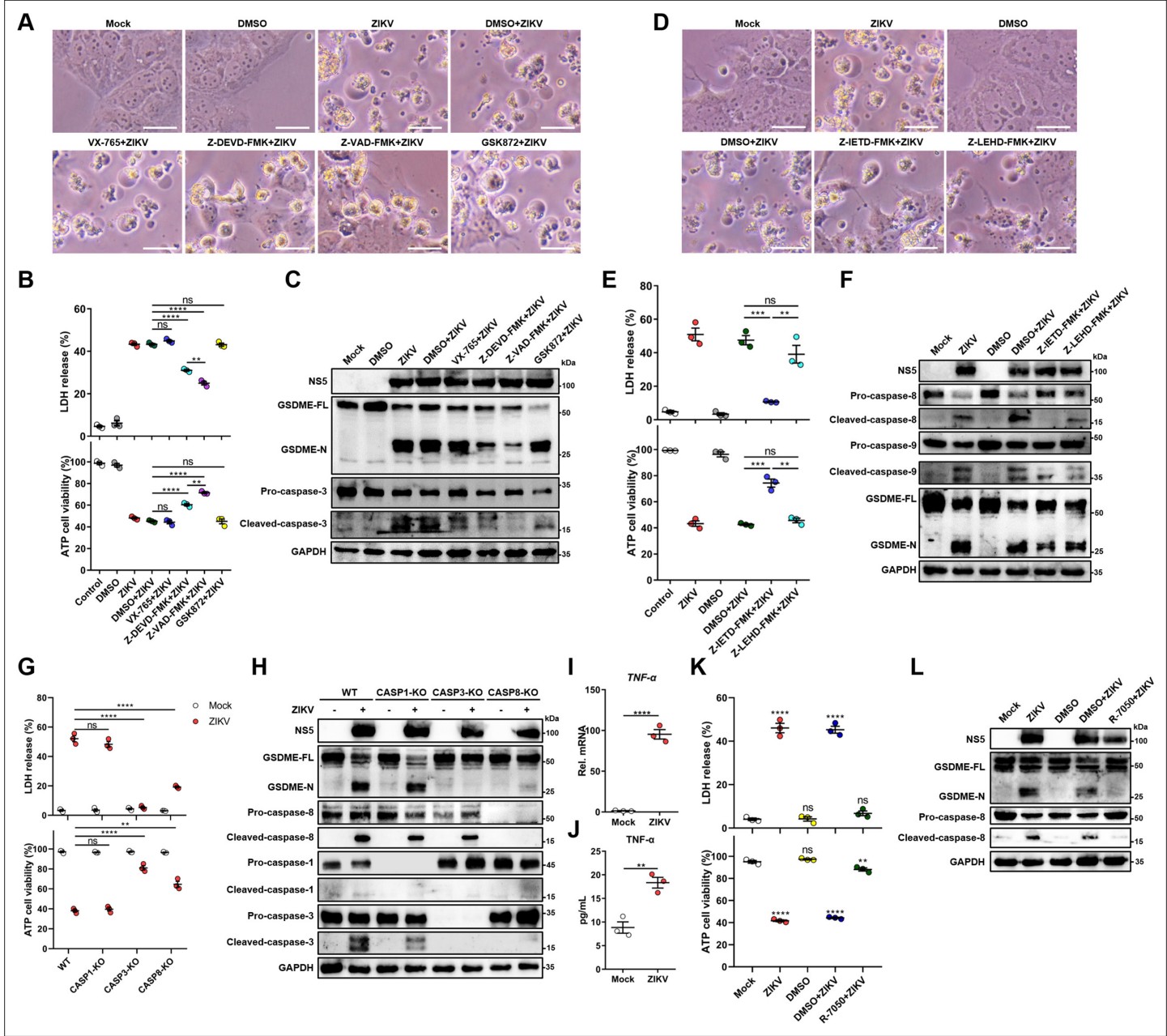

**Figure 3.** Zika virus (ZIKV) infection activates gasdermin E (GSDME) via extrinsic apoptotic pathway. (A–C) JEG-3 cells were infected with ZIKV at a multiplicity of infection (MOI) of 1 followed by incubated with either 10 μM VX-765, 25 μM Z-DEVD-FMK, 25 μM Z-VAD-FMK, or 10 μM GSK872. At 24 hr post-infection, the cells were subjected to microscopy (**A**), cytotoxicity (n=3) (**B**), and Western blot analyses (**C**). Scale bar, 50 μm. (D–F) JEG-3 cells were infected with ZIKV at an MOI of 1 followed by incubated with either 25 μM Z-IETD-FMK or 25 μM Z-LEHD-FMK. At 24 hr post-infection, the cells were subjected to microscopy (**D**), cytotoxicity (n=3) (**E**), and Western blot analyses (**F**). Scale bar, 50 μm. (G–H) Caspase-1, caspase-3, and caspase-8 KO JEG-3 cells or wild-type (WT) cells were infected with ZIKV at an MOI of 1. At 24 hr post infection, LDH levels in supernatant, cell viability (n=3) (**G**) and cleavage of GSDME, caspase-1, caspase-3, and caspase-8 were measured in ZIKV-infected JEG-3 cells. (I–J) JEG-3 cells were mock-infected or infected with ZIKV at an MOI of 1. At 24 hr post-infection, the mRNA level of *TNF-α* (**I**) and concentration of TNF-α in the culture supernatant of JEG-3 cells (**J**) (n=3) were determined by RT-qPCR and enzyme-linked immunosorbent assay (ELISA), respectively. (K–L) JEG-3 cells were infected with ZIKV at an MOI of 1 followed by incubated with 2 μM R-7050. At 24 hr post-infection, the cells were subjected to cytotoxicity (n=3) (**K**) and Western blot analyses (**L**). Unpaired t-test versus mock. All data are presented as the mean ± SEM of three independent experiments, and the two-tailed unpaired Student's t-test was used to calculate significance. **, p<0.01; ***, p<0.001; ****, p<0.0001; ns, no significance.

The online version of this article includes the following source data and figure supplement(s) for figure 3:

**Source data 1.** Raw data for *Figure 3*.

*Figure 3 continued on next page*

*Figure 3 continued*

**Figure supplement 1.** Zika virus (ZIKV) infection activates caspase-3 and caspase-8 in JEG-3 cells.

**Figure supplement 1—source data 1.** Raw data for *Figure 3—figure supplement 1*.

To further confirm our findings, caspase-1, caspase-3, and caspase-8 KO JEG-3 cell lines were generated individually, followed by infection with ZIKV. As shown in *Figure 3G and H* and *Figure 3—figure supplement 1E*, the deletion of caspase-3 completely abolished the ZIKV-induced GSDME-dependent pyroptosis, while the deletion of caspase-1 did not affect the ZIKV-induced activation of GSDME and LDH release. As expected, the KO of caspase-8 significantly inhibited ZIKV-induced pyroptosis and caspase-3 activation. These findings further verified that ZIKV induces pyroptosis by activating the caspase-8-caspase-3-GSDME axis.

Given that TNF-α is crucial for activating death receptor signaling and its downstream caspase-8, it is interesting to investigate whether TNF-α is involved in ZIKV-induced pyroptosis in JEG-3 cells. To this end, the production of TNF-α in ZIKV-infected or mock-infected JEG-3 cells was determined, which showed that ZIKV infection significantly promoted the production of TNF-α (*Figure 3I and J*). Then, ZIKV-infected JEG-3 cells were treated with TNF-α receptor antagonist (R-7050) and the GSDME-mediated pyroptosis was examined. The results showed that blockade of the TNF-α signaling pathway significantly inhibited the ZIKV-induced LDH release and the activation of caspase-8 and GSDME (*Figure 3K and L* and *Figure 3—figure supplement 1F*), suggesting that the induction of TNF-α expression contributes to ZIKV-induced pyroptosis. Taken together, these results demonstrated that ZIKV infection causes pyroptosis of JEG-3 cells via the TNF-α-caspase-8-caspase-3-GSDME signaling pathway.

## The genomic RNA of ZIKV activates the GSDME-dependent pyroptosis through the RIG-I-caspase-8-caspase-3 pathway

In order to explore which component of the ZIKV is responsible for inducing pyroptosis, JEG-3 cells were infected with ZIKV or UV-inactivated ZIKV to evaluate the contribution of viral structural proteins in inducing the pyroptosis. The results revealed that ZIKV structural proteins are not functional pyroptosis agonists (*Figure 4—figure supplement 1A and B*). Then JEG-3 cells were transfected with plasmid encoding individual viral NS proteins to determine the role of NS proteins in activating pyroptosis. However, the NS proteins still showed no effect on pyroptosis (*Figure 4—figure supplement 1C and D*), hence guiding us to speculate that the viral RNA may participate in orchestrating this phenomenon.

For this purpose, 5′ and 3′ untranslated region (UTR) RNAs of the ZIKV were obtained through the in vitro transcription assay to mimic the viral genome, followed by transfection of JEG-3 cells. As shown in *Figure 4A and B*, both UTRs induced the distinct pyroptotic cell death and LDH release, and compared to 3′ UTR, the 5′ UTR caused relatively a stronger effect on inducing pyroptosis (*Figure 4A and B*), hinting that the recognition of specific viral RNA motifs may aid the GSDME-dependent pyroptosis. Furthermore, the Western blot assay demonstrated that the ZIKV UTRs caused the activation of GSDME via the same signaling pathway as induced by ZIKV infection (*Figure 4C*).

Upon infection of the host cell by RNA viruses, cellular pattern recognition receptors (PRRs) play a decisive role in detecting viral RNAs and restricting viruses' replication. Among the PRRs, the retinoic acid-inducible gene I (RIG-I) and Toll-like receptor (TLR) 7 and 8 are responsible for sensing the viral ssRNA genome (*Hornung et al., 2006*; *Heil et al., 2004*). Therefore, whether RIG-I, TLR7, or TLR8 contributes to the onset of this was evaluated. The RIG-I-knockdown (KD), TLR7-KD, or TLR8-KD JEG-3 cell lines were generated using the CRISPR/Cas9 system (*Figure 4—figure supplement 2A*). ZIKV infection of engineered or wild-type (WT) cells revealed that the KD of RIG-I, but not TLR7 and TLR8, conspicuously attenuated the ZIKV-induced pyroptotic cell death and GSDME cleavage (*Figure 4D-F*), suggesting that RIG-I may play a major role in ZIKV-induced pyroptosis. To further verify the contribution of RIG-I in ZIKV-induced pyroptosis, RIG-I, MDA5, or MAVS KO JEG-3 cells were generated. As shown in *Figure 4G and H*, the deletion of RIG-I significantly inhibited the LDH release and the activation of GSDME, as well as its upstream caspase-3 and caspase-8 during ZIKV infection, while the KO of MDA5 or MAVS only slightly suppressed this process. Similarly, we also found that the deletion of RIG-I suppressed the lytic cell death and the activation of GSDME caused

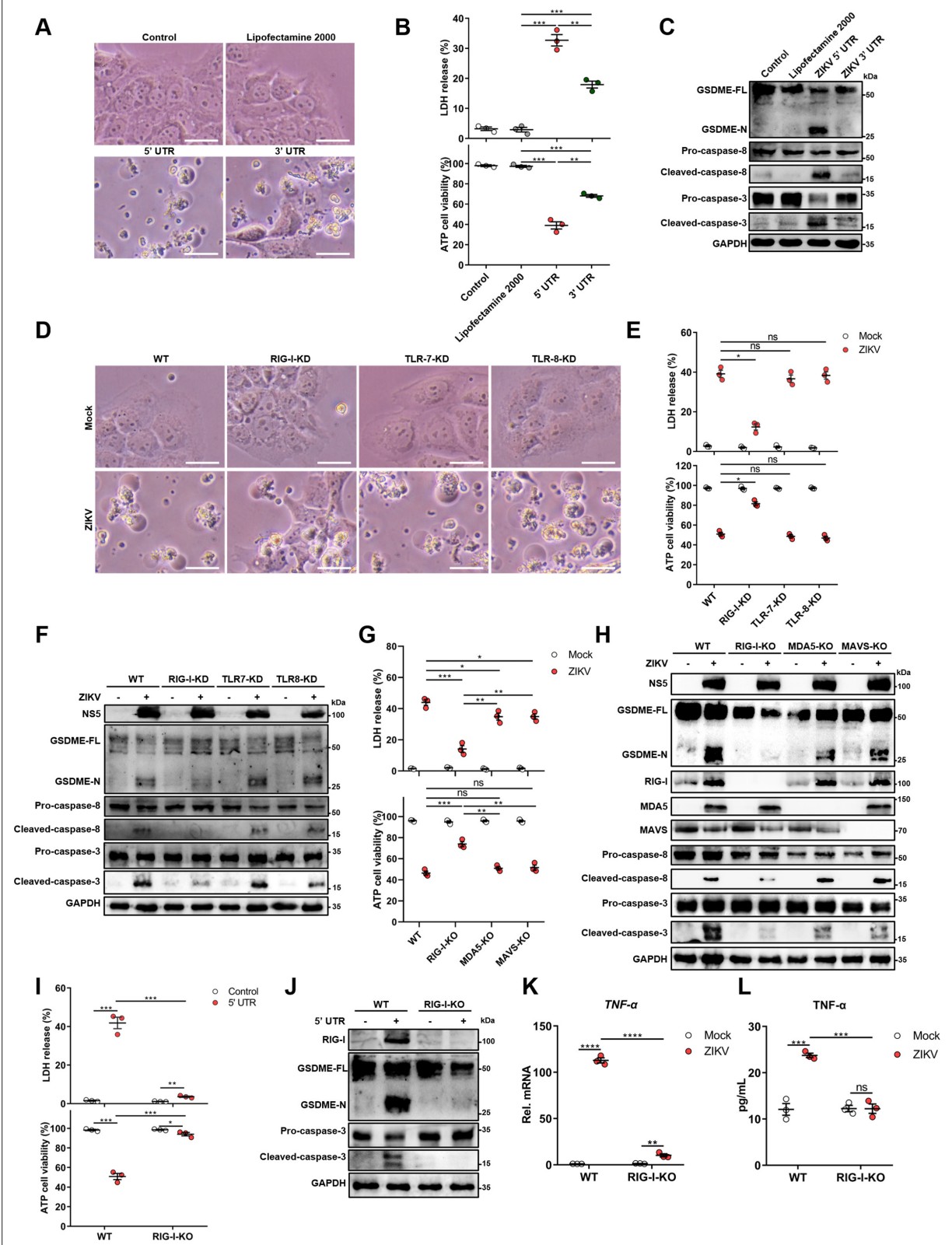

**Figure 4.** The genomic RNA of Zika virus (ZIKV) activates the gasdermin E (GSDME)-dependent pyroptosis through the RIG-I-caspase-8-caspase-3 pathway. (A–C) JEG-3 cells were seeded in six-well plates and were transfected with 1 μg ZIKV 5' untranslated region (UTR) or 3' UTR. At 24 hr post-transfection, cells were subjected to microscopy (**A**), cytotoxicity (n=3) (**B**), and Western blot analyses (**C**). Scale bar, 50 μm. (D–F) RIG-I, TLR7, or TLR8 knockdown JEG-3 cells or wild-type (WT) JEG-3 cells were infected with ZIKV at a multiplicity of infection (MOI) of 1. At 24 hr post-infection, cells were

*Figure 4 continued on next page*

*Figure 4 continued*

subjected to microscopy (**D**), cytotoxicity (n=3) (**E**), and immunoblot analyses (**F**). Scale bar, 50 μm. (**G–H**) ZIKV-infected JEG-3 cells were infected with ZIKV. At 24 hr post-infection, the cells were subjected to cytotoxicity (n=3) (**G**) and Western blot analyses (**H**). (**I–J**) JEG-3 cells and RIG-I KO JEG-3 cells were seeded in six-well plates and were transfected with 1 μg ZIKV 5′ UTR. At 24 hr post-transfection, cells were subjected to cytotoxicity (n=3) (**I**) and Western blot analyses (**J**). (**K–L**) JEG-3 or RIG-I KO JEG-3 cells were mock-infected or infected with ZIKV at an MOI of 1. The mRNA level (**K**) and concentration of TNF-α in the culture supernatant (**L**) (n=3) was determined by RT-qPCR and enzyme-linked immunosorbent assay (ELISA), respectively. All data are presented as the mean ± SEM of three independent experiments, and the two-tailed unpaired Student's t-test was used to calculate significance. **, p<0.01; ***, p<0.001; ****, p<0.0001; ns, no significance. RIG-I, retinoic acid-inducible gene I; TLR, Toll-like receptor.

The online version of this article includes the following source data and figure supplement(s) for figure 4:

**Source data 1.** Raw data for *Figure 4*.

**Figure supplement 1.** The structural and non-structural proteins of Zika virus (ZIKV) are not capable of inducing pyroptosis in JEG-3 cells.

**Figure supplement 1—source data 1.** Raw data for *Figure 4—figure supplement 1*.

**Figure supplement 2.** Activation of RIG-I is sufficient to induce gasdermin E (GSDME)-dependent pyroptosis.

**Figure supplement 2—source data 1.** Raw data for *Figure 4—figure supplement 2*.

by ZIKV 5′ UTR (*Figure 4I and J*). These results manifest that the ZIKV genome acts as a pivotal factor in inducing pyroptosis through the RIG-I pathway, and raising the possibility that GSDME-dependent pyroptosis can be launched as long as RIG-I is activated. In order to test this hypothesis, JEG-3 cells were transfected with poly(I:C) (RIG-I agonist) or human RIG-IN (the activation form of RIG-I) construct, and it was found that both poly(I:C) and RIG-IN could activate the lytic cell death and the cleavage of GSDME (*Figure 4—figure supplement 2B and C*), indicating that the activation of RIG-I is sufficient to induce GSDME-dependent pyroptosis.

Given the important role of RIG-I in mediating type I interferon (IFN-I) signaling, we then investigated whether IFN-I signaling contributed to the ZIKV-induced activation of GSDME. JEG-3 cells were treated with a polyclonal antibody against IFNAR1 to block IFN-I signaling, followed by transfection with ZIKV 5′ UTR. As shown in *Figure 4—figure supplement 2D and E*, blocking IFN-I signaling did not affect the 5′ UTR-induced GSDME activation, suggesting that IFN-I response may not be necessary for the activation of ZIKV-induced pyroptosis. To determine whether RIG-I mediated pyroptosis during ZIKV infection is mediated by TNF-α signaling, the production of TNF-α in ZIKV-infected RIG-I KO cells was determined. The results showed that the deletion of RIG-I prominently inhibited the ZIKV-induced production of TNF-α (*Figure 4K and L*), suggesting that RIG-I signaling could promote the production of TNF-α in ZIKV-infected cells. Taken together, our findings demonstrated that RIG-I-mediated recognition of the ZIKV genome could lead to pyroptosis of JEG-3 cells through activating TNF-α expression and caspase-8-caspase-3 signaling.

## ZIKV infection of pregnant immunocompetent C57BL/6N mice results in GSDME-mediated placental pyroptosis and CZS

To further verify the role of GSDME on ZIKV-induced placental damage and adverse fetal outcomes in vivo, a pregnant mouse model was employed. Before the in vivo experiments, mouse primary trophoblast cells (MTCs) from C57BL/6N mice at embryonic day 9.5 (E9.5) of pregnancy were isolated and cultured to investigate whether ZIKV infection could also induce the GSDME-mediated pyroptosis in mouse trophoblast cells (*Figure 5—figure supplement 1A*). As shown in *Figure 5—figure supplement 1B and C*, ZIKV infection significantly promoted the LDH release and the GSDME cleavage in MTCs. Consistent with our previous observations in JEG-3 cells, the upregulation of RIG-I caused by ZIKV infection was also seen in MTC, and inhibition of either caspase-3 or caspase-8 prominently blocked ZIKV-induced activation of GSDME (*Figure 5—figure supplement 1C*). These findings suggest that ZIKV activates pyroptosis through the same pathway in MTCs as that in JEG-3 cells.

Subsequently, WT or *Gsdme*[-/-] female mice were mated with male mice of their respective genotypes. Then, the pregnant mice were infected intravenously with 1×10[6] PFU of ZIKV H/PF/2013 strain or with an equal volume of Vero cell culture supernatant at E9.5 of pregnancy. The mice were sacrificed at E16.5 and the morphological appearance of placentas and individual fetuses was observed (scheme outlined in *Figure 5A*). Notably, although no clinical signs of disease and weight loss were observed in both ZIKV-infected WT and *Gsdme*[-/-] dams (*Figure 5—figure supplement 2A*), most of the WT dams infected with ZIKV showed abnormal pregnancies (*Figure 5—figure supplement 2B*,

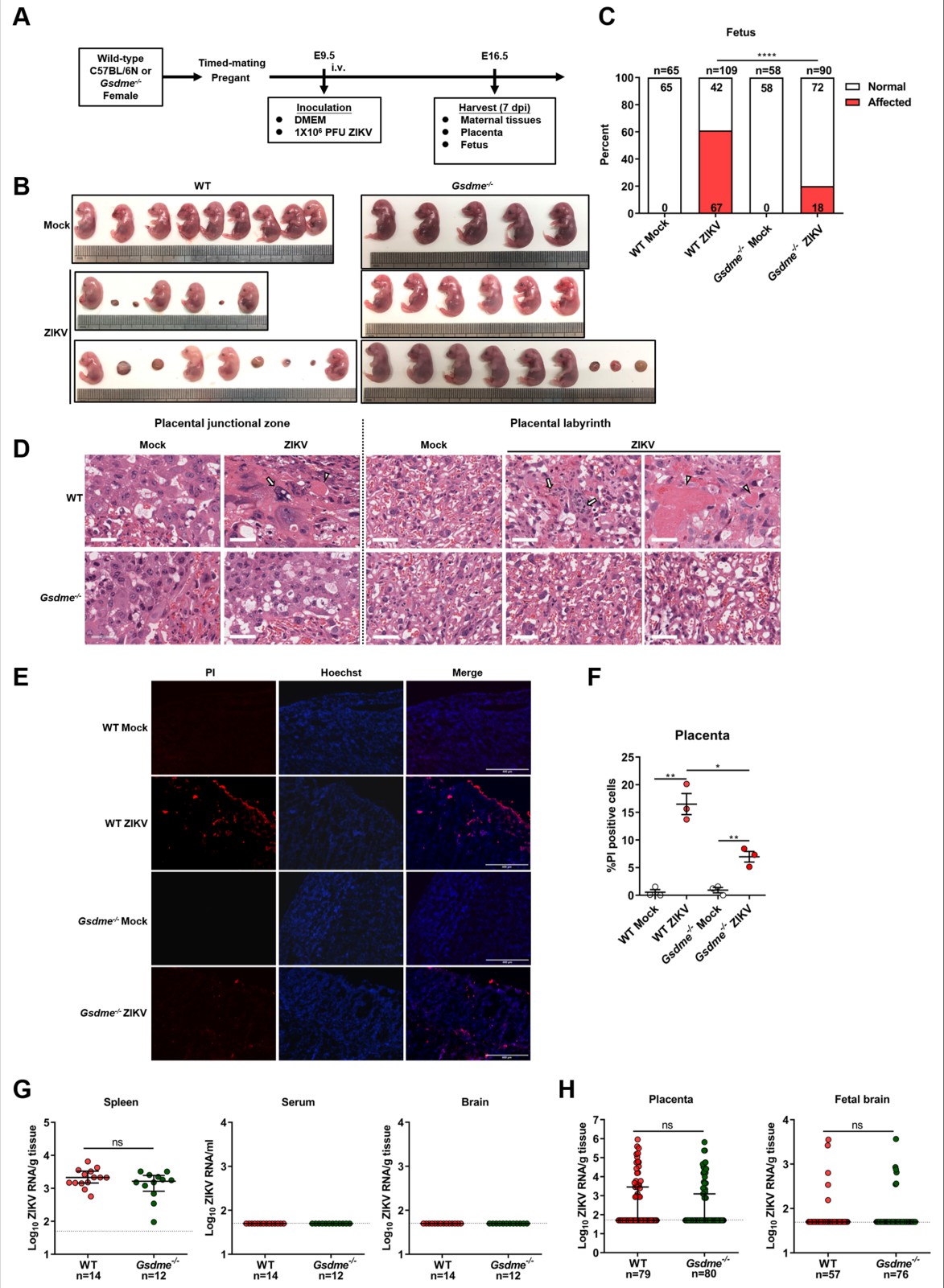

**Figure 5.** Zika virus (ZIKV) infection of pregnant immunocompetent wild-type (WT) C57BL/6N mice results in gasdermin E (GSDME)-mediated placental pyroptosis and congenital ZIKV syndrome (CZS). (**A**) Scheme of infection and the follow-up analyses. The WT or *Gsdme*⁻/⁻ female mice were mated with male mice of their respective genotypes, and the pregnant mice were infected intravenously with 1×10⁶ PFU of ZIKV H/PF/2013 strain or with an equal volume of Vero cell culture supernatant at embryonic day 9.5 (E9.5) of pregnancy. The mice were sacrificed at E16.5 and the placentas and

*Figure 5 continued on next page*

*Figure 5 continued*

individual fetuses were collected for follow-up experiments. (**B**) Representative images of fetuses from mock- and ZIKV-infected WT and *Gsdme*⁻/⁻ dams at E16.5 are shown. (**C**) Impact of ZIKV infection on fetuses from WT and *Gsdme*⁻/⁻ dams at E16.5. The percentage of fetuses that were affected (i.e., had undergone resorption, or exhibited any sign of growth restriction, or malformation) is shown. Numbers on bars indicate normal fetuses (top) or affected fetuses (bottom). (**D**) Representative hematoxylin and eosin staining showed pathological features of placentas at E16.5. Arrows indicate necrotic trophoblast cells. Arrowheads indicate thrombi. Scale bar, 50 µm. (**E–F**) Propidium iodide (PI) was intravenously injected into the mice before scarification. Representative placenta section images are shown (**E**) and PI-positive cells were quantified (n=3) (**F**). Scale bar, 400 µm. (**G**) ZIKV RNA levels of maternal spleens, serum, and brains of WT and *Gsdme*⁻/⁻ dams infected with ZIKV. (**H**) ZIKV RNA levels of all placentas and fetal heads carried by WT and *Gsdme*⁻/⁻ dams infected with ZIKV. The number of samples in each group is listed. Data for all panels are pooled from three to five independent experiments. For (**C**), significance was determined by Fisher's exact test. For (**F**), significance was determined by a two-tailed Student's t-test. For (**G**) and (**H**), the Mann-Whitney test was used to calculate significance. Data shown are median with interquartile range, and the dotted line depicts the limit of detection. *, p<0.05; **, p<0.01; ****, p<0.0001; ns, no significance.

The online version of this article includes the following source data and figure supplement(s) for figure 5:

**Source data 1.** Raw data for *Figure 5*.

**Figure supplement 1.** Zika virus (ZIKV) infection induces the gasdermin E (GSDME)-mediated pyroptosis in mouse primary trophoblast cells (MTCs).

**Figure supplement 1—source data 1.** Raw data for *Figure 5—figure supplement 1*.

**Figure supplement 2.** Gasdermin E (GSDME) deletion alleviates Zika virus (ZIKV)-induced abnormal pregnancy and adverse fetal outcomes.

**Figure supplement 2—source data 1.** Raw data for *Figure 5—figure supplement 2*.

left panel), and their fetuses showed a variable degree of CZS (*Figure 5B*, left panel). Among all 14 ZIKV-infected WT dams, only 2 dams displayed normal pregnancies. The remaining 12 dams exhibited abnormal pregnancies, with 4 dams containing only placental residues and embryonic debris as all fetuses had undergone resorption. Other 8 infected dams carried at least one or more placental residues and morphologically abnormal fetuses. Out of 109 implantation sites carried by all 14 infected dams, 67 (61%) sites were affected, among which 52 showed fetal resorptions and 15 showed growth restriction and malformation (*Figure 5C* and *Figure 5—figure supplement 2C*, left panel). These data indicate that ZIKV infection indeed leads to CZS in our experimental model. Although ZIKV-infected *Gsdme*⁻/⁻ dams also showed a certain degree of abnormal pregnancies, the number and severity of affected dams and fetuses were markedly decreased compared to the WT dams (*Figure 5B, C*, and *Figure 5—figure supplement 2B–2C*, right panel). Among all 10 ZIKV-infected *Gsdme*⁻/⁻ dams, only 4 dams showed abnormal pregnancies with a mixture of placental residues and stunted embryonic growth. The case in which all fetuses were resorbed in a dam vanished, and all fetuses carried by the rest of 6 dams showed no morphological abnormalities. The affected fetuses in the ZIKV-infected *Gsdme*⁻/⁻ dams were only 18 (20%), which is much lower than that in ZIKV-infected WT dams, Among the 18 affected fetuses, 14 showed fetal resorptions and 4 showed growth restriction and malformation. Thus, these data indicate that GSDME is associated with ZIKV-induced adverse fetal outcomes.

Furthermore, to evaluate the pathological changes associated with ZIKV infection, a general histological examination of the placentas was conducted, regardless of whether their corresponding fetuses were affected or not. As expected, the placentas of mock-infected WT mice presented normal features of the embryo-derived junctional and labyrinth zone, a place where the nutrient exchange occurs (*Figure 5D*). While in placentas of ZIKV-infected WT dams, obvious abnormal morphological changes, including hyperplasia, necrosis, and thrombi, were observed. Specifically, the hyperplastic trophoblast labyrinth showed denser cellularity and less vascular spaces, suggesting that the placental and embryonic blood circulation may have been compromised. Otherwise, abnormal spheroid structures were found in the junctional and labyrinth zone, which contained necrotic trophoblast cells, and thrombi were observed in the labyrinth zone. However, these pathological changes were not observed in the placentas from ZIKV-infected *Gsdme*⁻/⁻ dams (*Figure 5D*), suggesting that GSDME is essential to ZIKV-induced placenta pathology.

To further verify the role of GSDME on placental cell pyroptosis in ZIKV-infected mice, a propidium iodide (PI) staining assay was conducted in consideration of the fact that pyroptotic cells are permeable to small molecular weight, membrane-impermeable dyes such as 7-aminoactinomycin D, ethidium bromide, and PI (*Jorgensen and Miao, 2015*). No PI-positive cells were visualized in the placentas of mock-infected dams, while PI-positive signals were detected in the decidua and labyrinth zone in the placentas of ZIKV-infected WT dams. However, the placentas from ZIKV-infected *Gsdme*⁻/⁻ dams

showed relatively fewer PI-positive signals, implying that GSDME is involved in ZIKV-induced pyroptosis of placentas in vivo (*Figure 5E and F*).

In order to rule out the possibility that the variable infection status of ZIKV in WT and *Gsdme⁻/⁻* mice may confer variable fetal outcomes, we measured the ZIKV RNA levels in maternal tissues (spleen, brain, and serum), placentas, and fetuses (fetal brain). Comparable ZIKV RNA levels were detected in the spleens of these two types of mice, but the viral RNA remained undetectable in their brains and serum (*Figure 5G*). No significant differences were found in viral copies and infection ratio between the placentas and fetal brains from ZIKV-infected WT and *Gsdme⁻/⁻* dams (*Figure 5H*). Thereby, these data demonstrated that maternal GSDME-mediated placental pyroptosis promotes CZS without affecting ZIKV replication.

## Induction of TNF-α expression contributes to placental damage in ZIKV-infected pregnant mice

Since we found that TNF-α signaling contributes to pyroptosis in ZIKV-infected JEG-3 cells, we next explored whether this process is also involved in placental damage in ZIKV-infected pregnant mice. The expression and release of TNF-α in the placental tissue of ZIKV-infected or mock-infected mice was first measured by qRT-PCR and enzyme-linked immunosorbent assay (ELISA), respectively. Consistent with the in vitro result, ZIKV infection also significantly increased the production of TNF-α in mouse placenta (*Figure 6A and B*). To further determine the role of TNF-α signaling in ZIKV caused placental damage and adverse fetal outcomes, the pregnant C57BL/6N mice were intravenously infected with $1 \times 10^6$ PFU of ZIKV H/PF/2013 strain at E9.5, followed by treatment with R-7050 (7 mg/kg, i.p.) or DMSO every other day. At E16.5, placentas and individual fetuses were evaluated for morphological appearance. The results showed a significant reduction of affected fetuses in R-7050-treated mice compared to that in DMSO-treated mice, suggesting that the R-7050 treatment attenuated adverse fetal outcomes caused by ZIKV infection (*Figure 6C and D*, and *Figure 6—figure supplement 1*). Further histopathological analysis and PI staining assay showed that the R-7050 treatment effectively alleviated the placental damage and pyroptosis caused by ZIKV infection (*Figure 6E-G*). Collectively, our data demonstrate that the TNF-α signaling pathway plays an important role in ZIKV-induced placental damage and CZS, which highlights TNF-α as a potential therapeutic target for abnormal pregnancy and adverse fetal outcomes caused by ZIKV infection.

## Discussion

In this study, we identified a novel mechanism by which ZIKV infection induces the GSDME-mediated pyroptosis of placental cells and demonstrated that the KO of GSDME could alleviate the adverse fetal outcomes in ZIKV-infected mice. These data established a potential relevance of the placental pyroptotic process with the ZIKV-associated adverse fetal outcomes, thus suggesting that placental pyroptosis is a vital feature of ZIKV pathogenesis.

Previous studies have demonstrated that ZIKV infection activates inflammatory pathways and facilitates the secretion of pro-inflammatory cytokine interleukin-1β (IL-1β) in vitro and in vivo (*Tappe et al., 2016*; *He et al., 2018*). ZIKV NS5 and NS1 proteins, by interacting with NLRP3 and by stabilizing caspase-1, respectively, promote the NLRP3 inflammasome assembly/activation to augment the IL-1β secretion (*Wang et al., 2018*; *Zheng et al., 2018*). Since the NLRP3 inflammasome functions upstream of pyroptosis executor GSDMD and IL-1β is passively released during cell lysis (*Jorgensen and Miao, 2015*), it can be assumed that pyroptosis may confer ZIKV pathogenesis. Interestingly, our data revealed a previously unrecognized mechanism of ZIKV-induced fetal adverse outcomes, which is indirectly caused by GSDME-mediated pyroptosis. Unlike the GSDMD-mediated pyroptosis driven by caspase-1 and caspase-4/5/11, we found that GSDME undergoes activation by caspase-8 and caspase-3, which is partially consistent with published studies, wherein GSDME was subjected to be cleaved and activated by caspase-3 and granzyme B (*Shi et al., 2015*; *Kayagaki et al., 2015*; *Wang et al., 2017*; *Lu et al., 2018*). Recently, several studies have shown that GSDME-dependent pyroptosis is involved in the pathogenesis of viruses. For instance, the H7N9 influenza virus triggers GSDME-mediated pyroptosis in mouse alveolar epithelial cells, which in turn leads to a cytokine storm (*Wan et al., 2022*). GSDME is important for the pathogenesis of enterovirus 71 via mediating the initiation of pyroptosis (*Dong et al., 2022*). Also, there is evidence that GSDME-dependent pyroptosis is

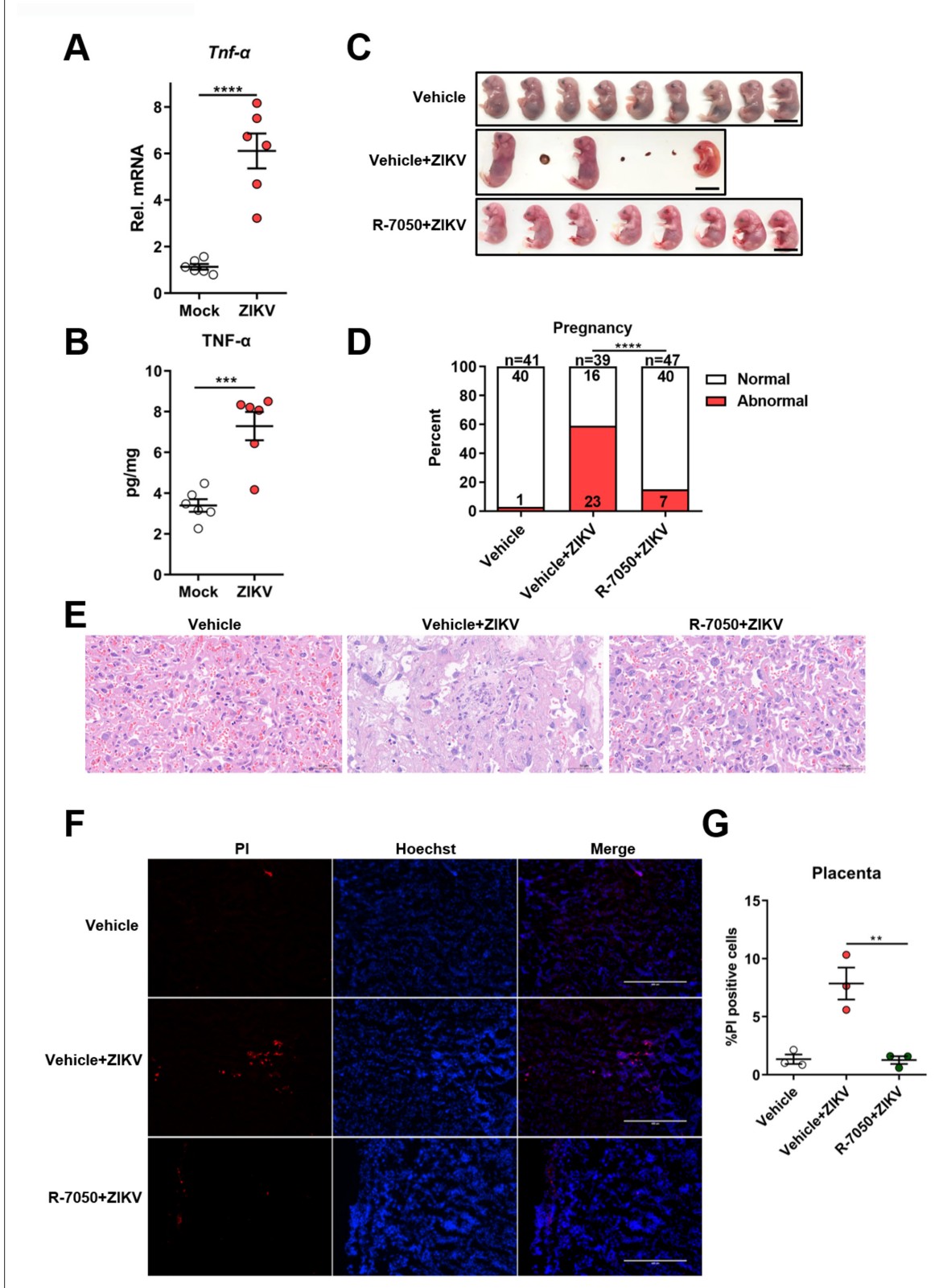

**Figure 6.** Induction of TNF-α expression contributes to placental damage in Zika virus (ZIKV)-infected pregnant mice. (A–B) The pregnant C57BL/6N mice were mock-infected or intravenously infected with 1×10⁶ PFU of ZIKV H/PF/2013 strain at embryonic day 9.5 (E9.5). At E16.5, placentas were collected and the mRNA level and concentration of TNF-α in mouse placentas were determined by RT-qPCR (**A**) and enzyme-linked immunosorbent assay (ELISA) (**B**), respectively (n=6). (C–G) The pregnant C57BL/6N mice were intravenously infected with 1×10⁶ PFU of ZIKV H/PF/2013 strain at E9.5,

*Figure 6 continued*

followed by treatment with R-7050 (7 mg/kg, i.p.) or DMSO every other day. At E16.5, mice were sacrificed and the placentas and individual fetuses were collected. Representative images of fetuses from mock- and ZIKV-infected dams treated with R-7050 or vehicle at E16.5 are shown. Scale bar, 1 cm (**C**). The percentages of fetuses that were affected (i.e., had undergone resorption, or exhibited any sign of growth restriction, or malformation) were calculated. Numbers on bars indicate normal fetuses (top) or affected fetuses (bottom) (**D**). Representative hematoxylin and eosin staining was performed to show the pathological features of placentas at E16.5. The asterisks indicate abnormal spheroid structure. Arrows indicate necrotic trophoblast cells. Arrowheads indicate thrombi. Scale bar, 50 μm (**E**). Propidium iodide (PI) was intravenously injected into the mice before scarification. Representative placenta section images are shown (**F**) and PI-positive cells were quantified (n=3) (**G**). Scale bar, 400 μm. For (**A**), (**B**), and (**G**) significance was determined by a two-tailed unpaired Student's t-test. For (**D**), significance was determined by Fisher's exact test. **, $p<0.01$; ***, $p<0.001$; ****, $p<0.0001$.

The online version of this article includes the following source data and figure supplement(s) for figure 6:

**Source data 1.** Raw data for *Figure 6*.

**Figure supplement 1.** R-7050 treatment attenuates Zika virus (ZIKV)-induced adverse fetal outcomes.

**Figure supplement 1—source data 1.** Raw data for *Figure 6—figure supplement 1*.

associated with the pathogenesis of SARS-CoV-2, as data has shown that the occurrence of pyroptosis in syncytia formed by fusion of SARS-CoV-2 spike and ACE2-expressing cells, which is initiated by the intrinsic apoptosis pathway and is executed by the caspase-3/7-mediated activation of GSDME (*Ma et al., 2021*). In addition to its role in viral pathogenesis, a study has also shown the indirect antiviral effect of GSDME. Precisely, the virus-mediated inactivation of anti-apoptotic Bcl-2 family members promotes caspase-3-dependent activation of GSDME in barrier epithelial cells, which restricts the virus replication (*Orzalli et al., 2021*). However, in our study, GSDME KO showed no effect on ZIKV replication both in vitro and in vivo, which excludes the possibility that the attenuated placental damage in ZIKV-infected GSDME KO mice is caused by the inhibited viral replication.

In our study, pyroptosis was not found as a universal phenomenon upon ZIKV infection, even though ZIKV can infect multiple cell lines. For instance, ZIKV-infected HeLa, HEK293T, and SH-SY5Y cells exhibited no apparent pyroptotic morphological changes, albeit some of them express GSDME in abundance. By evaluating the susceptibility of low GSDME-expressing HEK-293T and Huh-7 cells to ZIKV infection, we found that the infection susceptibility is an additional decisive factor, as Huh-7 cells, which are highly susceptible to ZIKV, were prone to pyroptosis (*Figure 2A-C* and *Figure 2—figure supplement 1*). These findings not only highlight the important role of GSDME in the pathogenesis of ZIKV, but also raise the question of whether other ZIKV-induced symptoms are associated with GSDME-mediated pyroptosis, which could be alluring to investigate in future studies.

Recent advances in understanding the key role of caspase-3 in mediating the GSDME-dependent pyroptosis have changed the long stand notion that the activation of caspase-3 is the hallmark of apoptosis (*Wang et al., 2017*). Through chemical inhibition and genetic KO, caspase-3 was identified to mediate the ZIKV-induced activation of GSDME, and further studies revealed that ZIKV activates caspase-3-GSDME-dependent pyroptosis through a caspase-8-dependent extrinsic apoptotic pathway. This conclusion is also supported by the observation that the blocking of TNF-α signaling inhibited ZIKV-induced pyroptosis in vitro and in vivo, as TNF-α has been known as a vital ligand to commence the caspase-8-mediated signaling (*Tummers and Green, 2017*). It is worth noting that caspase-3 can not only launch pyroptosis by activating GSDME but can also specifically block pyroptosis by cleaving GSDMD at a distinct site from the inflammatory caspases, although the conditions required for this process remain to be further explored (*Taabazuing et al., 2017*). In this study, the cleaved GSDMD-N was not detected in ZIKV-infected JEG-3 cells, which may be due to the different cell types used in our study. But in any case, the bidirectional crosstalk between apoptosis and pyroptosis is an issue that must be paid attention to in follow-up studies. Caspase-8 is a multifunctional effector protein related to various signaling pathways and acts as a molecular switch for apoptosis, necrosis, and pyroptosis (*Fritsch et al., 2019*). In the case of pyroptosis, caspase-8 is capable of inducing the GSDMD- and GSDME-mediated pyroptosis in murine macrophages (*Orning et al., 2018*; *Sarhan et al., 2018*; *Zeng et al., 2019*). However, our data showed that GSDMD was not activated by caspase-8 in ZIKV-infected JEG-3 cells, inferring the possibility that some specific factors are required for inducing the GSDMD-mediated pyroptosis. In addition, caspase-8 inhibits necrosis regulated by receptor-interacting serine/threonine kinase 3 (*Tummers and Green, 2017*), which fortifies our conclusion that necrosis did not occur in ZIKV-infected JEG-3 cells. Surprisingly, in

this study, caspase-8 inhibitor treatment also reduced the activation of caspase-9, and the caspase-9 inhibitor treatment slightly inhibited the activation of GSDME without any impact on the ZIKV-induced pyroptotic cell death (*Figure 3D-F*). Recent studies demonstrate that the GSDME-N can permeabilize the mitochondrial membrane and activate the intrinsic apoptotic pathway (*Rogers et al., 2019*), which implies the possibility that during ZIKV infection, the caspase-8-mediated activation of GSDME consequently induces the caspase-9-associated intrinsic apoptosis by targeting mitochondrial, thereby creating a self-amplifying feed-forward loop through consecutively activating caspase-3. Despite elusive mechanistic underpinnings, these data suggest that the extrinsic apoptotic pathway plays a key role in ZIKV-induced pyroptosis, and the potential crosstalk between pyroptosis and apoptosis during the ZIKV infection needs to be further explored.

During the viral infection of host cells, PRRs are vital for detecting the viral genomes and limiting the spread of viruses (*Brennan and Bowie, 2010*). Based on our findings that the ZIKV RNA genome, rather than the ZIKV-encoded proteins, spurs pyroptosis, we asked if PPRs, including TLR7, TLR8, and RIG-I, which are well-known sensors of single-strand RNA, were involved in the occurrence of this cell death process. Using the CRISPR-Cas9-engineered KO JEG-3 cells, we observed the necessity of RIG-I to induce the GSDME-mediated pyroptosis via recognizing the ZIKV RNA genome, with a piece of further evidence that the ZIKV 5' UTR was unable to launch the pyroptosis in RIG-I KO cells. These data are in concordance with a recent study which demonstrated that RIG-I recognizes the ZIKV 5' UTR in human cells and the enrichment of viral.

RNA on RIG-I decreases along with the genome toward the 3' UTR (*Chazal et al., 2018*), hence proposing a possibility that the activation of RIG-I determines the outcome of pyroptosis. This conclusion was subsequently confirmed by the finding that the cleavage of GSDME in JEG-3 cells could be activated by transfecting cells with human RIG-IN or RIG-I agonist poly(I:C). Interestingly, we found that KO of MAVS, a downstream molecule of RIG-I, only slightly reduced the cleavage of GSDME, which indicates the possibility that ZIKV infection leads to the placental pyroptosis via the non-canonical RIG-I pathway. Moreover, our results showed that the IFN signaling pathway is not essential for ZIKV-induced pyroptosis, which highlights the role of TNF-α in this process.

Since it has been well proven that murine IFN-I response restrains ZIKV infection, immunocompromised mice or WT mice treated with IFNAR1 antibodies were employed to establish animal models of ZIKV infection (*Miner et al., 2016*; *Szaba et al., 2018*). However, considering the fact that utilizing immunocompromised mice may prevent us from objectively assessing the potential impact of host innate immune responses on the pathogenesis of ZIKV and that the IFN signaling was found to play a vital role in mediating fetal demise after ZIKV infection by causing abnormal placental development (*Yockey et al., 2018*), the pregnant immunocompetent mouse model was employed in this study. This model has been successfully established in several studies, in which pregnant immunocompetent mice were infected with ZIKV by vaginal exposure or uterine infection, etc., and a definite association between adverse fetal outcomes and ZIKV infection in placenta and embryo was observed (*Yockey et al., 2016*; *Vermillion et al., 2017*). But considering the fact that these studies were done by exposing the placental-fetal interface to ZIKV and creating high viral loads in the vicinity of the placenta, which may not occur in the natural state of ZIKV infection. Therefore, in our study, the pregnant immunocompetent mice were infected with $1 \times 10^6$ PFU of ZIKV by tail vein injection, which is not lethal in WT and *Gsdme*$^{-/-}$ mice. Considering that ZIKV cannot replicate efficiently in immunocompetent mice, the dose we used here is much higher compared to other models using immunocompromised mice, but it is estimated to be within the range of the total amount of ZIKV that can be delivered by mosquito feeding (*Styer et al., 2007*; *Cox et al., 2012*). In our in vivo model, no clinical signs of disease and weight loss were observed in both ZIKV-infected WT and *Gsdme*$^{-/-}$ dams. This is consistent with the results of previous studies which revealed that ZIKV infection could not lead to any clinical symptoms in immunocompetent pregnant mice except placental damage and adverse fetal outcomes (*Szaba et al., 2018*; *Barbeito-Andrés et al., 2020*). This observation excludes the possibility that the effects of GSDME KO on the fetus is due to decreased pyroptosis in other tissues in the dams, but the use of conditionally deficient mice would make the conclusion more convincing.

Although the current data do support the notion that GSDME is involved in ZIKV pathogenesis, our study still exhibits some limitations. First, there is a dearth of effective antibodies to detect the cleavage of GSDME in vivo. Although we used the *Gsdme*$^{-/-}$ mice model to indirectly substantiate that the GSDME ablation significantly inhibited the ZIKV-induced adverse fetal outcomes and demonstrated

that ZIKV could activate GSDME in mouse trophoblast primary cells, the direct evidence for ZIKV-induced activation of placental GSDME in vivo yet needs to be assessed. Moreover, the observed CZS in our study is whether contributed by the maternal and fetal GSDME remains unclear. Given that a study, which also employed the intravenously infected immunocompetent mice, as objects have shown that the placental pathology rather than the embryonic/fetal viral infection could be a stronger contributor to adverse pregnancy outcomes in mice (*Szaba et al., 2018*), we speculate that GSDME in the maternal placenta may play a more crucial role in this process. This speculation is indirectly supported by another recent study, which employed subcutaneously infected 3-day-old mouse pups as an experimental model and has demonstrated that the inhibition of caspase-3 did not affect ZIKV-induced neuropathology and brain atrophy in vivo (*He et al., 2020*). This finding implies the possibility that caspase-3- and GSDME-dependent pyroptosis is not involved in fetal brain damage due to direct ZIKV infection of the mouse brain. Ultimately discerning the maternal versus fetal contributions of GSDME would be beneficial by crossing the *Gsdme*$^{+/-}$ and *Gsdme*$^{-/-}$ mice and by comparing pathology in +/- and -/- littermates. However, considering that the placenta is formed by the joint participation of the mother and the embryo, it is a challenge to accurately answer this question in vivo.

It is worth noting that regardless of whether placental lesions and fetal resorption occurred or not, the ZIKV RNA was infrequently detected, which suggests that placental function may be affected under a negligible quantity of viral particles or the virus had been eliminated at this stage. As the placenta acts as a crucial tissue between the gravida and embryo, preventing pathogen transmission during pregnancy and mediating the exchange of nutrients imply that any inappreciable placental damage can lead to devastating consequences. In addition, the ZIKV-infected *Gsdme-/-* dams still showed a certain degree of CZS, albeit the proportion of affected embryos carried by them was strikingly lower than the WT dams, highlighting that the GSDME is not the only decisive factor to regulate the ZIKV-induced pyroptosis.

Our findings in understanding the key role of TNF-α in the pathogenesis of ZIKV prompted us to explore new therapeutic strategies against ZIKV. R-7050 is a cell-permeable triazoloquinoxaline inhibitor of the TNF-α receptor complex, which has been shown to exert a therapeutic effect in multiple murine disease models, including kidney fibrosis, intracerebral hemorrhage, and ischemic stroke (*Wen et al., 2019*; *King et al., 2013*; *Lin et al., 2021*). Also, it has been revealed that the R-7050 treatment inhibits the activation of caspase-8 and caspase-3 by blocking the TNF-α signaling pathway (*Pal et al., 2021*). Our data revealed that the R-7050 treatment effectively alleviates ZIKV-induced placental damage and adverse fetal outcomes in the pregnant mouse model. Although the clinical safety and effectiveness of R-7050 need to be further tested, our study provides a potential new therapeutic strategy to overcome CZS due to ZIKV infection.

In summary, our study identified a novel mechanism of ZIKV-caused placental damage and CZS. We demonstrated that ZIKV infection could induce placental pyroptosis through the recognition of the viral genome by RIG-I followed by TNF-α release, which activates the caspase-8- and caspase-3-mediated cleavage of pyroptosis executor GSDME in placental cells. Furthermore, the ablation of GSDME or the treatment of TNF-α receptor inhibitor in infected pregnant mice attenuated placental pyroptosis and adverse fetal outcomes. These findings may provide opportunities to establish therapies that might halt placental damage and subsequent adverse fetal outcomes during ZIKV infection.

## Materials and methods
### Reagents and antibodies
Mouse monoclonal anti-ZIKV NS5 antibody was generated in our lab. Rabbit monoclonal anti-GSDME (ab215191) antibody was purchased from Abcam. Mouse monoclonal anti-GAPDH (AC002), mouse monoclonal anti-Flag (AE005), rabbit monoclonal anti-TLR7 (A19126), rabbit polyclonal anti-TLR8 (A12906), rabbit monoclonal anti-caspase-3 (A19654) antibodies, rabbit monoclonal anti-CK7 (A4357), rabbit monoclonal anti-MDA5 (A2419), and rabbit polyclonal anti-MAVS (A5764) were purchased from ABclonal Technology. Rabbit polyclonal anti-GSDMD (96458), mouse monoclonal anti-caspase-8 (9746), and rabbit polyclonal anti-caspase-9 (9502) antibodies were purchased from Cell Signaling Technology. Rabbit polyclonal anti-RIG-I (20566-1-AP) was purchased from Proteintech. Rabbit polyclonal anti-IFNAR1 (13222-T20) antibody was purchased from SinoBiological. Horseradish

peroxidase-labeled goat anti-mouse (AS003) or anti-rabbit (AS014) secondary antibodies were obtained from ABclonal Technology.

The chemical reagents DAPI (D8417), PI (P4170), and Hoechst (B2261) were purchased from Sigma-Aldrich. The inhibitors specific to pan-caspase (zVAD-FMK, S7023), caspase-1 (VX-765, S2228), caspase-3 (Z-DEVD-FMK, S7312), caspase-8 (Z-IETD-FMK, S7314), caspase-9 (Z-LEHD-FMK, S7313), RIP3K (GSK-872, S8465), and TNF-α receptor (R-7050, S6643) were purchased from Selleck.

## Cell lines, viruses, and plaque assay

Human embryonic kidney (HEK-293T, CRL-11268), human alveolar epithelial adenocarcinoma (A549, CCL-185), African green monkey kidney (Vero, CCL-81) Aedes albopictus mosquito (C6/36, CRL-1660), human cervical cancer (HeLa, CCL-2), human neuroblastoma (SH-SY5Y, CRL-2266), and human choriocarcinoma/trophoblastic cancer (JEG-3, HTB-36) cell lines were obtained from the American Type Culture Collection (ATCC) and verified by ATCC. Human hepatocellular carcinoma (Huh-7) cell line was stored in our lab and matched the STR reference profile of Huh-7. HEK-293T, A549, Huh-7, and Vero cells were cultured in Dulbecco's modified Eagle's medium (DMEM; Sigma) supplemented with 100 U/ml penicillin, 100 g/ml streptomycin, and 10% fetal bovine serum (FBS, Gibco). C6/36 and HeLa cells were cultured in RPMI-1640 medium (Hyclone) supplemented with 10% FBS and 2 mM L-glutamine. SH-SY5Y and JEG-3 cells were grown in DMEM supplemented with 10% FBS and MEM Non-essential Amino Acid Solution (Sigma). C6/36 cells were grown at 28°C in a 5% $CO_2$ incubator, whereas all other cells were grown at 37°C in a 5% $CO_2$ incubator. All cell lines were routinely tested for mycoplasma by PCR kit (Beyotime Biotechnology) and found negative.

The ZIKV strain H/PF/2013 (Accession: KJ776791.2) was obtained from Wuhan Institute of Virology, Chinese Academy of Sciences, and propagated in Vero cells and was titrated by plaque assays on Vero cells. In brief, relevant cells were inoculated with ZIKV at an MOI of 0.1, and the culture supernatants were harvested at 12–72 hr post-infection. The supernatants were then serially diluted and inoculated onto the monolayers of Vero cells. After 1 hr of absorption, cells were washed with serum-free DMEM and cultured in DMEM containing 3% FBS and 1.5% sodium carboxymethyl cellulose (Sigma-Aldrich). Visible plaques were counted, and viral titers were calculated after 5 days of incubation.

## Isolation of MTCs

A detailed description of the procedure used to isolate trophoblasts has been reported in a previous publication (*Zhou et al., 2008*). Briefly, C57BL/6N mice were set up for timed matings, and at E9.5, the fetoplacental unit was separated from uterine implantation sites, and washed with pre-cooling HBSS solution (Sigma). The placentas were removed from the fetoplacental unit and cut into small pieces (~1 mm³). The obtained tissues were digested at 37°C in four cycles of 10 min by 0.25% trypsin (Gibco), which contains 0.02% DNase type I (Beyotime Biotechnology), 25 mM HEPES (Thermo Fisher Scientific) and 4.2 mM $MgSO_4$ (Beyotime Biotechnology), with gentle vibration. After each digestion, the cell suspension was filtered through a 40 µm cell strainer (Corning). Subsequently, the filtered cells were collected and layered over a discontinuous Percoll Gradient (65–20%, in 5% steps), and centrifuged at 1000× g for 20 min. The cells at densities between 1.048 and 1.062 g/ml were collected, and washed three times with DMEM/F12 medium. Cell counts were then performed, and the cells were diluted to 1×10⁶ cells/ml and maintained in DMEM/F12 medium containing 10% FBS (Gibco). Finally, cells were seeded in 12-well plates, which were pre-coated with Matrigel (Corning), and incubated at 37°C in a 5% $CO_2$ incubator. The rabbit monoclonal anti-CK7 (Abcloanl Technology, A4357) antibody was used as marker for cells of trophoblast lineage.

## Cytotoxicity assay

Cell death was assessed by the LDH assay using the CytoTox 96 Non-Radioactive Cytotoxicity Assay kit (Promega, G1780). Cell viability was determined by the CellTiter-Glo Luminescent Cell Viability Assay kit (Promega, G7570). All the relevant experiments were performed following the manufacturer's instructions.

## Western blot analysis

Total cell lysates were prepared using the RIPA buffer (Sigma, R0278) containing protease inhibitors (Roche, 04693116001). After sonication, the protein concentration in each sample was determined by

using the BCA protein assay kit (Thermo Fisher Scientific) and boiled at 95°C for 10 min. Equivalent amounts of protein samples were separated by SDS-PAGE and electroblotted onto a polyvinylidene fluoride membrane (Roche) using a Mini Trans-Blot Cell (Bio-Rad). The membranes were blocked at room temperature (RT) for 2 hr in PBS containing 3% bovine serum albumin (BSA), followed by incubation with the indicated primary antibodies overnight at 4°C. After washing three times with TBS-Tween (50 mM Tris-HCl, 150 mM NaCl, and 0.1% [v/v] Tween 20, pH 7.4), the membranes were incubated with secondary antibodies at RT for 45 min. Finally, the membranes were visualized with a chemiluminescence system (Tanon) after three times of wash.

## Microscopy and immunofluorescence analysis

To examine the morphology of pyroptotic cells, relevant cells were seeded in six-well plates at ~40% confluency and then subjected to indicated treatments. Phase-contrast cell images were captured by the NIKON Ti-U microscope.

For immunofluorescence analysis, relevant cells were seeded in 12-well plates and infected with ZIKV at an MOI of 0.1. At indicated time post-infection, cells were fixed with pre-cold methanol for 10 min at −20°C. Subsequently, cells were washed three times by PBS and were blocked in PBS containing 1% BSA for 1 hr at RT. Thereafter, cells were incubated with indicated primary antibody for 2 hr at RT, followed by washing three times with PBS and incubation with Alexa Fluor 488-conjugated goat anti-mouse (Invitrogen, A-11001) or Alexa Fluor 488-conjugated goat anti-rabbit (Invitrogen, A-11008) for 45 min. Cells were then washed and incubated with DAPI (Sigma, D8417) for another 10 min at RT. After final washing, cells were visualized using the EVOS FL auto (Thermo Fisher Scientific).

## Plasmid construction and transfection

Using the ZIKV cDNA as template, NS1, NS2A, NS2B3P, NS3H, NS4A, NS4B, and NS5 were separately cloned into the p3XFLAG-CMV-10 using the PCR/restriction digest-based cloning method, and were finally verified by sequencing. Transfections were performed using the Lipofectamine 2000 (Invitrogen) according to the manufacturer's instructions.

## CRISPR-Cas9 KD and KO cells

Two guide RNAs (gRNAs) targeted to each desired gene were cloned into the lentiviral vector lenti-CRISPR v2. About 800 ng lentiviral vector, 400 ng packaging plasmid pMD2.G, and 800 ng pSPAX2 were co-transfected into HEK 293T cells using the FuGENE HD Transfection Reagent (Promega) according to the manufacturer's instructions. At 48 hr post-transfection, viral supernatants were collected, and then inoculated to $4 \times 10^5$ JEG-3 cells for another 48 hr. For RIG-I, TLR-7, and TLR-8 KD in JEG-3 cells, the gRNA-expressing cells were selected with 1.5 µg/ml puromycin, and then plated into 12-well plates for further experiments. For GSDME, caspase-1, caspase-3, caspase-8, and RIG-I KO in JEG-3 cells, the gRNA-expressing cells were selected with 1.5 µg/ml puromycin following the limiting dilution analysis. The single-cell clones were cultured in 96-well plates for another 10 days or longer. The immunoblotting was used to screen for GSDME-deficient clones and to verify the KD efficiency. The genotyping of the KO cells was determined by sequencing.

The sequences of gRNAs were as follows: gRNA-*GSDME*-1, 5′-TAAGTTACAGCTTCTAAGTC-3′; gRNA-*GSDME*-2, 5′-CAGTTTTTATCCCTCACCCT-3′; gRNA-*RIG-I*-1, 5′-GGATTATATCCGGAAGA CCC-3′; gRNA-*RIG-I*-2, 5′-TCCTGAGCTACATGGCCCCC-3′; gRNA-*TLR7*-1, 5′-GGTGAGGTTCGTG GTGTTCG-3′; gRNA-*TLR7*-2, 5′-CCTGCGGTATCTCTAGTAGC-3′; gRNA-*TLR8*-1, 5′-GTGCAGCAA TCGTCGACTAC-3′; gRNA-*TLR8*-2, 5′-AATCCCGGTATACAATCAAA-3′; gRNA-*CASP1*-1, 5′-AAGGA TATGGAAACAAAAGT-3′; gRNA-*CASP1*-2, 5′-CCACATCCTCAGGCTCAGAA-3′; gRNA-*CASP3*-1, 5′-ACTAATATAAACAGAAGGCG-3′; gRNA-*CASP3*-2, 5′-GAATGGCACAAACATTTGAAA-3′; gRNA-*CASP8*-1, 5′-CTACCTAAACACTAGAAAGG-3′; gRNA-*CASP8*-2, 5′-GCCTGGACTACATTCCGCAA-3′; gRNA-*MDA5*-1, 5′-CGTCTTGGATAAGTGCATGG-3′; gRNA-*MDA5*-2, 5′-GCGTTCTCAAACGATGG AGA-3′; gRNA-*MAVS*-1, 5′-CTGTGAGCTAGTTGATCTCG-3′; gRNA-*MAVS*-2, 5′-TCTTCAATACCCT TCAGCGG-3′.

## Mice and infections

C57BL/6N WT mice were purchased from Vital River Laboratory Animal Technology Co. *Gsdme*$^{-/-}$ mice were kindly provided by Prof. Feng Shao (National Institute of Biological Sciences, Beijing). Mice used

for all experiments were naive. No drug tests were done. Mice were housed under 12/12 hr light/dark cycle and up to five animals per cage, with access to food and water ad libitum. Eight-week-old mice randomly selected and set up for timed matings, and pregnant dams were injected intravenously with 50 µl Vero cell culture supernatant or $1 \times 10^6$ PFU of ZIKV in a volume of 50 µl at the E9.5. All pregnant dams were sacrificed at E16.5, and all of the placentas, fetuses, and maternal tissues were harvested for the subsequent analyses.

## Measurement of viral burden

ZIKV-infected pregnant mice were euthanized at E16.5. Maternal tissues, fetuses, and their corresponding placentas were harvested. Samples were weighed and homogenized with stainless steel beads in 1 ml of DEME supplemented with 2% heat-inactivated FBS. All homogenized tissues were stored at –80°C until virus titration was performed. Some placentas were divided equally into two parts, and one part was fixed in 4% paraformaldehyde for histological examination.

Total RNA in homogenized samples and serum were extracted using the TRIzol Reagent (Invitrogen) according to the manufacturer's instructions, and were reverse-transcribed to cDNA with random hexamers. ZIKV RNA levels were determined by quantitative real-time PCR on a 7500 Real-time PCR system (Applied Biosystems). Viral burden was expressed on a log10 scale as viral RNA equivalents per g or per ml after comparison with a standard curve produced using the serial 10-fold dilutions of ZIKV RNA. The ZIKV-specific primer set and probe were used as published previously (*Lanciotti et al., 2008*): forward primer, 5'-CCGCTGCCCAACACAAG-3'; reverse primer, 5'-CCACT AACGTTCTTTTGCAGACAT-3'; and Probe, 5'-/56-FAM/AGCCTACCT/ZEN/TGACAAGCAATCAG ACACTCAA/3IABkFQ/–3' (Sangon).

## Histological and immunohistochemistry staining

For histological staining, placentas were fixed in 4% paraformaldehyde overnight at 4°C and embedded in paraffin. At least three placentas from different litters were sectioned and stained with hematoxylin and eosin.

For immunohistochemistry staining, ~5 µm thick paraffin sections were placed in 3% $H_2O_2$ for 30 min to quench the endogenous peroxidase activity, and then the sectioned slides were incubated in citrate buffer at 96°C for 30 min for antigen retrieval. After washing with PBS containing 0.1% Tween 20, the sections were blocked in 5% BSA for 1 hr and incubated overnight at 4°C with rabbit anti-cleaved-caspase-3 primary antibody (Servicebio) diluted in PBS with 0.1% Tween 20. After washing, the sections were incubated with secondary antibody (horseradish peroxidase-labeled sheep anti-rabbit IgG, Beijing ZSGB-BIO Co., Ltd) for 45 min. Finally, the slides were developed using the 3,3'-diami-nobenzidine, and hematoxylin was used for counterstaining. All immunohistochemical sections were scanned with a Leica Apero CS2 slide scanning system.

## In vivo PI staining assay of placental pyroptosis

Mock- or ZIKV-infected mice were administered PI (1.5 mg/kg) via the intravenous route at E16.5 before subjecting them to sacrifice. About 15 min later, mice were euthanized, and the placentas were collected and embedded in the OCT compound (SAKURA, 4583). After slicing, the slides were stained with Hoechst and scanned on EVOS FL auto (Thermo Fisher Scientific).

## In vitro RNA transcription and transfection

The DNA fragments corresponding to the 5' UTR and 3' UTR of ZIKV genome were cloned using the cDNAs from viral stocks. The 5' and 3' UTR fragments were amplified by PCR using the following primers: 5' UTR forward primer 5'-CGGGGTACCAGTTGTTGATCTGTGTGAATCAGA-3', 5' UTR reverse primer 5'-CCGCTCGAGGACCAGAAACTCTCGTTTCCAAA-3', 3' UTR forward primer 5'- CGGGGTACCGCA CCAATCTTAGTGTTGTCAGGCC-3', and 3' UTR reverse primer 5'- CCGCTCGAGAGACCCATGGA TTTCCCCAC-3'. Amplified fragments were digested with KpnI and XhoI and cloned into the pre-cut pcDNA4/myc-His A vector (Invitrogen) by using the DNA ligation Kit (Takara). Plasmid DNA was then linearized with the restriction enzyme XhoI, and used as a template for T7 in vitro transcription using the mMESSAGE mMACHINE T7 Transcription Kit (Ambion). The RNA was precipitated with lithium chloride and quantified by spectrophotometry. Transfections were performed using the Lipofectamine 2000 (Invitrogen) according to the manufacturer's instructions.

## RNA extraction and quantitative real-time PCR

Total RNA in treated cells was extracted using TRIzol Reagent (Invitrogen), and 1 μg RNA was used to synthesize cDNA using a first-strand cDNA synthesis kit (TOYOBO). Quantitative real-time PCR was performed using a 7500 Real-Time PCR System (Applied Biosystems) and SYBR Green PCR Master Mix (TOYOBO). Data were normalized to the level of *ACTB or Actb* expression in each sample. The primer pairs used were as follows: human *ACTB*: forward 5'-AGCGGGAAATCGTGCGTGAC-3', reverse 5'-GGAAGGAAGGCTGGAAGAGTG-3'; human *TNF-α*: forward 5'-CCTCTCTAATCAGCCCT CTG-3', reverse 5'-GAGGACCTGGGAGTAGATGAG-3'; mouse *Actb*: forward 5'-CACTGCCGCATCC TCTTCCTCCC-3', reverse 5'- CAATAGTGATGACCTGGCCGT-3'; mouse *Tnf-α*: forward 5'-CTCAG CCTCTTCTCATTCCTGC-3', reverse 5'-GGCCATAGAACTGATGAGAGGG-3'.

## Enzyme-linked immunosorbent assay

The cell supernatant was collected and centrifuged at 1000× *g* for 10 min at 4°C to remove particulates. The concentration of TNF-α in the cell supernatant was determined by human TNF-α ELISA kit (Abclonal Technology, RK00030) according to the manufacturer's instructions. For mouse placenta tissue, the placentas were cut into pieces (~1 mm$^3$) and homogenized in pre-cooling HBSS solution (Sigma) on ice. The tissue suspension was then sonicated with an ultrasonic cell disrupter till the solution is clarified, and centrifuged at 10,000× *g* for 10 min at 4°C. Collect the supernatant for subsequent experiments. The concentration of TNF-α was determined by mouse TNF-α ELISA kit (Abclonal Technology, RK00027) according to the manufacturer's instructions.

## Statistical analysis

All data were analyzed with the GraphPad Prism software. For viral burden analysis, the log titers and levels of viral RNA were analyzed by nonparametric Mann-Whitney tests. Contingency data were analyzed by Fisher's exact test using numbers in each group. The statistical analysis of differences between two groups was performed using two-tailed unpaired Student's t-test. For all statistical significance indications in this manuscript, ****, $p < 0.0001$; ***, $p < 0.001$; **, $p < 0.01$; *, $p < 0.05$ and ns, no significance.

## Acknowledgements

We acknowledge F Shao (National Institute of Biological Sciences, Beijing, China) for providing *Gsdme*[-/-] mice. National Key Research and Development Program of China 2021YFC2600200, National Natural Science Foundation of China 32022082, National Natural Science Foundation of China 31825025, National Natural Science Foundation of China 31972721, National Natural Science Foundation of China 32030107, Natural Science Foundation of Hubei Province 2019CFA010.

## Additional information

### Funding

| Funder | Grant reference number | Author |
| --- | --- | --- |
| National Key Research and Development Program of China | 2021YFC2600200 | Shengbo Cao |
| National Natural Science Foundation of China | 32022082 | Jing Ye |
| National Natural Science Foundation of China | 31825025 | Shengbo Cao |
| National Natural Science Foundation of China | 31972721 | Jing Ye |
| National Natural Science Foundation of China | 32030107 | Shengbo Cao |

| Funder | Grant reference number | Author |
|---|---|---|
| Natural Science Foundation of Hubei Province | 2019CFA010 | Shengbo Cao |

The funders had no role in study design, data collection and interpretation, or the decision to submit the work for publication.

## Author contributions

Zikai Zhao, Data curation, Software, Formal analysis, Validation, Investigation, Methodology, Writing – original draft; Qi Li, Mengjie Yang, Wenjing Zhu, Jun Gu, Zheng Chen, Changqin Gu, Investigation; Usama Ashraf, Youhui Si, Writing – review and editing; Shengbo Cao, Conceptualization, Supervision, Funding acquisition, Methodology, Project administration, Writing – review and editing; Jing Ye, Conceptualization, Supervision, Funding acquisition, Project administration, Writing – review and editing

## Author ORCIDs

Jing Ye http://orcid.org/0000-0002-3258-6224

## Ethics

All animal studies were conducted in strict accordance with the Guide for Care and Use of Laboratory Animals of Laboratory Animal Centre, Huazhong Agriculture University, and all experiments conform to the relevant regulatory standards. The experiments and protocols were approved by the Animal Management and Ethics Committee of Huazhong Agriculture University (Assurance number HZAUMO-2020-0053).

## Decision letter and Author response

Decision letter https://doi.org/10.7554/eLife.73792.sa1
Author response https://doi.org/10.7554/eLife.73792.sa2

# Additional files

## Supplementary files

• MDAR checklist

## Data availability

All data generated or analyzed during this study are included in the manuscript and supporting file.

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
