## [Editor Report]

The study of Zika virus-induced cell death is critical for our understanding of viral pathogenesis. The authors have studied the molecular mechanisms by which Zika virus causes cell death.

---

## [Decision Letter]

**Decision letter after peer review:**

Thank you for submitting your article "Zika virus causes placental pyroptosis and associated adverse fetal outcomes by activating GSDME" for consideration by *eLife*. Your article has been reviewed by 3 peer reviewers, and the evaluation has been overseen by a Reviewing Editor and Tadatsugu Taniguchi as the Senior Editor. The reviewers have opted to remain anonymous.

Essential revisions:

1. Two reviewers mentioned that the pathway analysis on the role of caspases is a particularly weak part of this study. Genetic analysis of (at a minimum) casp3, casp8 and casp1 (as control) should be performed using the CRISPR KO approach used in Figure 1 for GSDME. The chemical inhibitor data present on casp3 and casp8 in Figure 3 is not sufficient and confusing. The authors conclude that casp8 is required for Zika induced pyroptosis, but there is no effect of casp8 inhibition on GSDME cleavage (Figure 3H). Only with genetic KO cells can this pathway be mapped. This should be addresses with new experiments.

2. An exciting part of the study is the interesting but surprising claim that the ZIKV genome UTRs are sufficient to induce GSDME pyroptosis in a RIG-I dependent manner. To my knowledge, no such pathway has been described. Hence, a few additional experiments to shore up that observation are merited. For example, one prediction is that other RIG-I agonists should similarly induce pyroptosis. Moreover, RIG-I, MDA5, and MAVS knockouts would allow for clear, genetic determination of whether or not the pathway in question operates similarly to known RIG-I signaling. This was mentioned by two reviewers and should be addresses with new experiments.

3. Essential details on the mouse studies are lacking especially because more conventional models are in the IFNAR KO background. Given that the authors describe pyroptosis in other cell types, it seems possible that the effects of GSDME knockout on the fetus could be indirect and due to decreased pyroptosis in elsewhere in the dams. How did GSDME knockout alter the clinical signs of disease (weight loss, histopathology) in the dams? It would be helpful if the authors contextualized their model relative to those previously published. There was also a concern that the mechanistic studies were performed in one cancer cell line (JEG3 and that the upstream pathway is not addressed at all in the in vivo model (e.g., experiments using Casp3 or RigI KOs)). Another significant issue with the in vivo work is that the mating scheme does not allow us to discern the maternal vs. fetal contributions of GSDME. Although ideally this should be addressed with new experiments, we realize that this could cause unnecessarily delay of publication. Therefore, we ask to better describe the current data and explicitly detail the limitations of the in vivo work in the discussion of the revised manuscript.

*Reviewer #1 (Recommendations for the authors):*

In the study "Zika virus causes placental pyroptosis and associated adverse fetal outcomes by activating GSDME", the authors report that Zika infection induces GSDME mediated cell death. The significance of this finding is high, as the impact of Zika on cell death in the placenta is probably key to pathogenesis. The data implicating GSDME in vitro and in vivo is impressive. However, the pathways proposed that lead to GSDME cleavage and pyroptosis are not supported by the data. Much additional work is needed here. If this pathway is mapped as suggested, the impact on the field would be significant.

1. The casp3 western blots presented throughout the study are unimpressive and difficult to interpret. As casp3 can be released from dead cells, it is possible that the westerns may be easier to interpret if cell lysates and supernatants are examined for cleaved and full length casp3. These studies are suggested.

2. The pathway analysis on the role of caspases is the weakest part of this study. Genetic analysis of (at a minimum) casp3, casp8 and casp1 (as control) should be performed using the CRISPR KO approach used in Figure 1 for GSDME. The chemical inhibitor data present on casp3 and casp8 in Figure 3 is not sufficient and confusing. The authors conclude that casp8 is required for Zika induced pyroptosis, but there is no effect of casp8 inhibition on GSDME cleavage (Figure 3H). Only with genetic KO cells can this pathway be mapped.

3. The role of TNF-induced casp8 activity as a driver of GSDME cleavage and pyroptosis is not compelling. The authors are using cells that are very weak inducers of TNF expression, and the only data presented here on a role for TNF is qPCR of this cytokine (not ELISA). I am convinced that GSDME is involved here, but the upstream regulation needs significant analysis.

4. The authors are strongly encouraged to consider a recent study (PMID: 33979579), which established GSDME as a regulator of virus-induced pyroptosis (as opposed to a role in cancer). That study showed that mitochondrial disruption during viral infection activates casp3 to cleave GSDME and promote cell death. The authors of this current study also see a role for casp3. Casp8 was not involved. Based on the weak evidence for a role of TNF and casp8 in GSDME cleavage induced by Zika, the authors are encouraged to consider alternative models.

5. The role of RIG-I in this process should be considered in the context of the fact that RIG-I can promote the activation of PKR, which shuts down translation (PMID: 25891073). As the study cited in point 4 above showed that translation shutdown activates casp3-GSDME, it is possible that the RIG-I dependent phenotype observed here is explained by this prior work. The authors should assess if RIG-I mediated IFNs are important for pyroptosis or if RIG-I mediated PKR activation is important by assessing the role of MAVS and PKR by CRISPR KO.

6. Line 328 states: "researches on GSDME have mainly focused on its anti-cancer effects, but little is known about its function in the pathogenesis of virus. Our research not only identifies for the first time that GSDME can be activated by ZIKV, but also hints the important role of GSDME in viral pathogenesis."

While correct that this is the first report of a role for GSDME in Zika infection, authors should cite the prior work on GSDME with other viruses.

*Reviewer #2 (Recommendations for the authors):*

– Figure 4: This is perhaps the most surprising, interesting, and potentially contentious result. To my knowledge, RNA sensing leading to a RIG-I>CASP8>CASP3>GSDME pathway has not been described, and the mechanistic link from RIG-I to CASP8 is unclear. I do not think that the authors need to fully map out this pathway (e.g., the link between RIG-I and CASP8), but I do think that additional experiments are required to shore up the key observation that RIG-I is the ZIKV sensor upstream of GSDME pyroptosis. Thus, experiments in Figure 4 should be repeated using knockouts (instead of knockdowns) for RIG-I, MDA5, and MAVS. Well known agonists of RIG-I should also be tested to determine if this pathway activates GSDME pyroptosis in JEG-3 cells, or if there is something specific to the ZIKV genome that activates a 'non-canonical' RIG-I pathway.

– A surprising omission is an experiment to address the effect of GSDME pyroptosis on viral replication. Although there is not an overt in vivo phenotype on ZIKV replication, viral titers from ZIKV infected WT and GSDME-/- JEG-3 cells should be addressed. This is important for interpreting whether GSDME pyroptosis is only driving pathogenesis, or if GSDME pyroptosis may also play a role in viral clearance.

– Based on the mating scheme, it is not feasible to discern the maternal vs. fetal contributions of GSDME. This is a key question and a limitation of the study design and current results. This should at a minimum be acknowledged in the Discussion.

– The in vivo data significantly increases the impact of the manuscript, and I think is a major factor in considering the manuscript for publication in *eLife*. Although I strongly prefer not to request additional in vivo experiments, the authors currently provide no data to demonstrate that the pathway in mouse trophoblasts is the same as observed in the human JEG-3 cells. Some evidence demonstrating that murine GSDME is being activated by ZIKV via the same pathway as shown for human trophoblasts should be included. Perhaps the most rigorous demonstration would be to test if Rig-I or Casp3 knockout mice phenocopy Gsdme knockouts. However, I would be satisfied if the same pathway was rigorously defined in a mouse trophoblast cell line or primary cells using knockouts or inhibitors.

*Reviewer #3 (Recommendations for the authors):*

Given the results in Figure 4, could RIG-I expression in the cell lines in Figure 2 explain differences in pyroptosis?

Weight measurements of fetuses would be preferable to the "affected" scoring system in Figure 5C and 6B (which is not described).

---

## [Author Response]

Essential revisions:1. Two reviewers mentioned that the pathway analysis on the role of caspases is a particularly weak part of this study. Genetic analysis of (at a minimum) casp3, casp8 and casp1 (as control) should be performed using the CRISPR KO approach used in Figure 1 for GSDME. The chemical inhibitor data present on casp3 and casp8 in Figure 3 is not sufficient and confusing. The authors conclude that casp8 is required for Zika induced pyroptosis, but there is no effect of casp8 inhibition on GSDME cleavage (Figure 3H). Only with genetic KO cells can this pathway be mapped. This should be addresses with new experiments.

Thanks for the comments. To show the chemical inhibitor data more clearly, a grayscale analysis of the Western blot results was performed. As shown in Figure 3—figure supplement 1B and 1D, compared to untreated and caspase9 inhibitor-treated cells, caspase8 inhibitor-treated cells showed significant suppression of GSDME cleavage. Moreover, we constructed caspase1, caspase3, and caspase8 knock-out (KO) JEG-3 cell lines individually according to the reviewers’ suggestion. As shown in Figure 3G and 3H, caspase1 KO did not affect the ZIKV-induced LDH release and activation of GSDME, while the deletion of caspase3 completely abolished the ZIKV-induced GSDME-dependent pyroptosis. Although ZIKV-infected caspase8 KO cells still showed a certain degree of pyroptosis, this process was significantly inhibited compared to ZIKV-infected wild-type cells. Taken together, these results demonstrated the important roles of caspase3 and caspase8 in ZIKV-induced pyroptosis, and further confirmed that the caspase1-GSDMD signaling pathway is not involved in ZIKV-induced pyroptosis in JEG-3 cells.

2. An exciting part of the study is the interesting but surprising claim that the ZIKV genome UTRs are sufficient to induce GSDME pyroptosis in a RIG-I dependent manner. To my knowledge, no such pathway has been described. Hence, a few additional experiments to shore up that observation are merited. For example, one prediction is that other RIG-I agonists should similarly induce pyroptosis. Moreover, RIG-I, MDA5, and MAVS knockouts would allow for clear, genetic determination of whether or not the pathway in question operates similarly to known RIG-I signaling. This was mentioned by two reviewers and should be addresses with new experiments.

Thanks for the comments. Our study did describe for the first time that ZIKV genome UTRs are sufficient to induce GSDME pyroptosis in a RIG-I-dependent manner. To make this conclusion more reliable, some additional experiments were carried out according to the reviewers’ suggestion. Firstly, the expression of RIG-I in ZIKV-infected and uninfected cells were detected by Western blot. As shown in Figure 4H and Figure 5—figure supplement 1C, ZIKV infection or 5’ UTR prominently upregulated the expression of RIG-I in JEG-3 cells and mouse primary trophoblast cells.

To explore whether activation of RIG-I can induce GSDME-dependent pyroptosis in the absence of ZIKV infection, JEG-3 cells were transfected with RIG-I agonist poly(I:C) or human RIG-IN construct (active mutant of RIG-I). As shown in Figure 4—figure supplement 2B and 2C, both poly(I:C) and RIG-IN induced the cleavage of GSDME, indicating that activation of RIG-I is sufficient to induce GSDME-dependent pyroptosis.

To further determine whether RIG-I mediates GSDME-dependent pyroptosis through the canonical RIG-I pathway, RIG-I, MDA5 or MAVS KO JEG-3 cells were generated, followed by ZIKV infection. As shown in Figure 4G and 4H, the deletion of RIG-I significantly inhibited the activation of GSDME, as well as its upstream caspase3 and caspase8. Although ZIKV-induced GSDME activation was also slightly inhibited in MDA5 or MAVS KO JEG-3 cell lines, the effect is much weaker than that in RIG-I KO JEG-3 cell line.

As RIG-I signaling has been found to be important for type I interferon (IFN-I) response during flavivirus infection, to further investigate whether IFN-I signaling contributed to the ZIKV-induced activation of GSDME, a polyclonal antibody against IFNAR1 was used to block IFN-I signaling. The results showed that blocking IFN-I signaling is not involved in ZIKV 5’UTR-induced activation of GSDME (Figure 4—figure supplement 2D and 2E).

Taken together, these results and our previous results demonstrate that RIG-I plays an important role in ZIKV-induced GSDME pyroptosis independent of IFN-I signaling.

3. Essential details on the mouse studies are lacking especially because more conventional models are in the IFNAR KO background. Given that the authors describe pyroptosis in other cell types, it seems possible that the effects of GSDME knockout on the fetus could be indirect and due to decreased pyroptosis in elsewhere in the dams. How did GSDME knockout alter the clinical signs of disease (weight loss, histopathology) in the dams? It would be helpful if the authors contextualized their model relative to those previously published. There was also a concern that the mechanistic studies were performed in one cancer cell line (JEG3 and that the upstream pathway is not addressed at all in the in vivo model (e.g., experiments using Casp3 or RigI KOs)). Another significant issue with the in vivo work is that the mating scheme does not allow us to discern the maternal vs. fetal contributions of GSDME. Although ideally this should be addressed with new experiments, we realize that this could cause unnecessarily delay of publication. Therefore, we ask to better describe the current data and explicitly detail the limitations of the in vivo work in the discussion of the revised manuscript.

Thanks for the comments. Since it is too difficult to generate *Ifnar^-/-^Gsdme^-/-^* mice, we can only choose to conduct in vivo experiments by using immunocompetent mice rather than IFNAR KO mice. The pregnancy models of ZIKV infection in immunocompetent mice have been well-established in the previous studies (Szaba et al., 2018), and the *Gsdme^-/-^* mice used in our study were kindly provided by Dr. F Shao whose study has demonstrated that deletion of GSDME doesn’t affect the development and immune system of mice (Wang et al., 2017). In our in vivo model, no clinical signs of disease and weight loss were observed in both ZIKV-infected WT and *Gsdme^-/^*^-^ dams. This is consistent with the results of previous studies which revealed that ZIKV-infection did not lead to any clinical symptoms in immunocompetent pregnant mice except placental damage and adverse fetal outcomes (Szaba et al., 2018; Barbeito-Andres et al., 2020). Our data showing extremely low viral loads in spleens, serum and brains of infected dams and no difference of viral titers in tissues between WT and *Gsdme^-/-^* dams (Figure 5G) also support that GSDME knockout did not alter the clinical signs of disease. The figure showing the weight change of dams (Figure 5—figure supplement 2A) and relative discussion were added in the revised manuscript. In addition, the majority of affected embryos underwent resorption, leaving only the placental residues and embryonic debris, so it’s hard to evaluate the function of GSDME by histopathological methods in embryos. In the remaining embryo samples, no obvious clinical signs were found in both WT embryos and *Gsdme^-/-^* embryos.

We also realized that our current in vivo data could not distinguish the function of GSDME on the placental side and fetal side. In fact, we crossed *Gsdme*^+/-^ and *Gsdme*^-/-^ mice and hoped to address this question by comparing pathology in +/- to -/- littermates. However, the pregnancy and litter rates were too low, even though we spent a lot of time and tried many times, we still could not get enough data to draw conclusions, so we added discussion of the limitations of our in vivo experiments in revised manuscript (Line 467-486).

Considering RIG-I-deficiency has been shown to affect the replication of flaviviruses due to its important role in innate immunity (Guo et al., 2018), we decided not to evaluate the function of GSDME in RIG-I KO mice. In addition, we could not obtain enough *Caspase*3^-/-^ mice in such a short time to conduct experiments. Therefore, to address the mechanism in the in vivo model, the role of TNF-α, which is known as the upstream regulator of caspase8 and was shown to be significantly upregulated in ZIKV-infected JEG-3 cells and mouse placentas, in ZIKV-induced placental pyroptosis was determined. To this end, we found that RIG-KO significantly reduced the expression of TNF-α induced by ZIKV infection (Figure 4K and 4L); blocking the TNF-α signaling by using R-7050 (a TNF-α receptor antagonist) restrained the activation of caspase8 and GSDME in ZIKV-infected JEG-3 cells (Figure 3K and 3L). Further in vivo experiments demonstrated that blocking the TNF-α signaling alleviated placental damage and inhibited abnormal pregnancy and adverse fetal outcomes caused by ZIKV infection (Figure 6). These findings suggest that ZIKV mediates the activation of caspase8 by promoting the expression of TNF-α through the RIG-I pathway, which eventually leads to the pyroptosis of placenta.

According to the reviewers’ suggestion, mouse primary trophoblast cells (MTC) were isolated followed by infection of ZIKV. As shown in Figure 5—figure supplement 1, ZIKV infection significantly upregulated the activation of GSDME and release of LDH in MTC, while inhibition of either caspase3 or caspase8 could markedly suppress this process. These findings suggest that ZIKV induces GSDME-dependent pyroptosis in mouse trophoblast cells through the same pathway as in human JEG-3 cells.

Reviewer #1 (Recommendations for the authors):In the study "Zika virus causes placental pyroptosis and associated adverse fetal outcomes by activating GSDME", the authors report that Zika infection induces GSDME mediated cell death. The significance of this finding is high, as the impact of Zika on cell death in the placenta is probably key to pathogenesis. The data implicating GSDME in vitro and in vivo is impressive. However, the pathways proposed that lead to GSDME cleavage and pyroptosis are not supported by the data. Much additional work is needed here. If this pathway is mapped as suggested, the impact on the field would be significant.1. The casp3 western blots presented throughout the study are unimpressive and difficult to interpret. As casp3 can be released from dead cells, it is possible that the westerns may be easier to interpret if cell lysates and supernatants are examined for cleaved and full length casp3. These studies are suggested.

Our western blot results clearly showed that ZIKV-infection could induce the cleavage of caspase3 (Figure 3—figure supplement 1A), while knock-out caspase8 or knock-down/out RIG-I obviously reduced the cleavage (Figure 3H, Figure 4F and 4H). To further evaluate the role of caspase3 in ZIKV-induced pyroptosis, the caspase3-KO JEG-3 cell line was generated followed by ZIKV infection, and the results showed that the deletion of caspase3 completely abolished the ZIKV-induced activation of GSDME, suggesting that caspase3-mediated activation of GSDME plays a crucial role in ZIKV infection (Figure 3G and 3H).

2. The pathway analysis on the role of caspases is the weakest part of this study. Genetic analysis of (at a minimum) casp3, casp8 and casp1 (as control) should be performed using the CRISPR KO approach used in Figure 1 for GSDME. The chemical inhibitor data present on casp3 and casp8 in Figure 3 is not sufficient and confusing. The authors conclude that casp8 is required for Zika induced pyroptosis, but there is no effect of casp8 inhibition on GSDME cleavage (Figure 3H). Only with genetic KO cells can this pathway be mapped.

According to the reviewer’s suggestion, caspase1, caspase3, and caspase8 KO JEG-3 cells were generated. As shown in Figure 3G and 3H in the revised manuscript, the ZIKV-infected caspase8-KO cells still showed a certain degree of pyroptosis; however, the activation of GSDME and caspase3 were significantly inhibited compared to those in infected wild-type cells, suggesting that caspase8 plays an important role in ZIKV-induced pyroptosis. In parallel, our results showed that caspase1-KO did not affect the activation of GSDME upon ZIKV infection, while this process was completely abolished in caspase3-KO JEG-3 cells. These results suggest that caspase8 and caspase3 are required for ZIKV-induced pyroptosis.

3. The role of TNF-induced casp8 activity as a driver of GSDME cleavage and pyroptosis is not compelling. The authors are using cells that are very weak inducers of TNF expression, and the only data presented here on a role for TNF is qPCR of this cytokine (not ELISA). I am convinced that GSDME is involved here, but the upstream regulation needs significant analysis.

Thanks for the comments. We did not examine whether the expression of TNF-α would be significantly altered by ZIKV infection in other cells, since we wanted to focus our study on placenta or placenta-derived cells. Actually, we measured the expression of many inflammatory cytokines, including IL-1β, IL-6, and TNF-α, and only the expression of TNF-α was significantly upregulated in ZIKV-infected JEG-3 cells. Similar results were also found in ZIKV-infected mouse placentas (Figure 3I, 3J, 6A, and 6B). To further verify the role of TNF-α, a TNF-α receptor antagonist, R-7050, was used. We found that blocking the TNF-α signaling pathway by R-7050 treatment can significantly inhibit the activation of caspase8 as well as its downstream caspase3 and GSDME (Figure 3K and 3L), indicating that the upregulated expression of TNF-α caused by ZIKV infection is critical for the activation of GSDME. In addition, as suggested by the reviewer, the production of TNF-α in cell culture and placenta tissue was determined by ELISA. In consistent with RT-qPCR results, ZIKV infection increased the protein concentration of TNF-α in the supernatant of JEG-3 cells (Figure 3I and 3J) and mouse placenta (Figure 6A and 6B), while RIG-I KO restored the release of TNF-α induced by ZIKV infection (Figure 4K and 4L).

4. The authors are strongly encouraged to consider a recent study (PMID: 33979579), which established GSDME as a regulator of virus-induced pyroptosis (as opposed to a role in cancer). That study showed that mitochondrial disruption during viral infection activates casp3 to cleave GSDME and promote cell death. The authors of this current study also see a role for casp3. Casp8 was not involved. Based on the weak evidence for a role of TNF and casp8 in GSDME cleavage induced by Zika, the authors are encouraged to consider alternative models.

In this study, the roles of both the exogenous apoptosis pathway and the endogenous apoptosis pathway upstream of caspase3 were investigated. The former refers to the permeabilization of the mitochondrial membrane and the assembly of apoptosome resulting in the activation of caspase-9, and the latter involves the activation of death receptors and caspase-8. By using the chemical inhibitors, we found that caspase8, but not caspase9, played a major role in regulating pyroptosis (Figure 3E, 3F, and Figure 3—figure supplement 1B). Moreover, our data showed that inhibiting the TNF-α signaling pathway by TNF-α receptor antagonist or caspase8 KO could significantly inhibit ZIKV-induced pyroptosis, which further affirmed the important role of TNF-α and caspase8 in ZIKV-induced pyroptosis (Figure 3G, 3H, 3K, and 3L). Therefore, we believe that our current data are sufficient to support the viewpoint that caspase8 plays a major role in ZIKV-induced pyroptosis. We also agree that the report (PMID: 33979579) is highly informative to us, so we have added a discussion about this article and issues related to virus-induced mitochondrial disruption in the revised manuscript (Line 378 to 381, and 415 to 425).

5. The role of RIG-I in this process should be considered in the context of the fact that RIG-I can promote the activation of PKR, which shuts down translation (PMID: 25891073). As the study cited in point 4 above showed that translation shutdown activates casp3-GSDME, it is possible that the RIG-I dependent phenotype observed here is explained by this prior work. The authors should assess if RIG-I mediated IFNs are important for pyroptosis or if RIG-I mediated PKR activation is important by assessing the role of MAVS and PKR by CRISPR KO.

In order to determine whether RIG-I mediates GSDME-dependent pyroptosis through the canonical RIG-I pathway, RIG-I, MDA5, or MAVS KO JEG-3 cells were generated. As shown in Figure 4G and 4H, the deletion of RIG-I significantly inhibited the activation of GSDME, as well as its upstream caspase3 and caspase8. On the contrary, ZIKV-induced GSDME activation was just slightly inhibited in MDA5 or MAVS KO JEG-3 cell lines. Furthermore, to assess whether RIG-I mediated IFNs signaling is involved in ZIKV-induced pyroptosis, a polyclonal antibody against IFNAR1 was used to block the IFN-I signaling, and it was found that blocking IFN-I signaling did not affect the ZIKV 5’UTR-induced activation of GSDME (Figure 4—figure supplement 2D and 2E). These results suggest that RIG-I mediated IFN signaling is not involved in ZIKV-induced pyroptosis, and rule out the potential role of PKR, an interferon-stimulated gene, in ZIKV-induced pyroptosis.

6. Line 328 states: "researches on GSDME have mainly focused on its anti-cancer effects, but little is known about its function in the pathogenesis of virus. Our research not only identifies for the first time that GSDME can be activated by ZIKV, but also hints the important role of GSDME in viral pathogenesis."While correct that this is the first report of a role for GSDME in Zika infection, authors should cite the prior work on GSDME with other viruses.

We have corrected the inappropriate description and cited the prior work on GSDME with other viruses in the Discussion section in the revised manuscript (Line 370 to 381).

Reviewer #2 (Recommendations for the authors):– Figure 4: This is perhaps the most surprising, interesting, and potentially contentious result. To my knowledge, RNA sensing leading to a RIG-I>CASP8>CASP3>GSDME pathway has not been described, and the mechanistic link from RIG-I to CASP8 is unclear. I do not think that the authors need to fully map out this pathway (e.g., the link between RIG-I and CASP8), but I do think that additional experiments are required to shore up the key observation that RIG-I is the ZIKV sensor upstream of GSDME pyroptosis. Thus, experiments in Figure 4 should be repeated using knockouts (instead of knockdowns) for RIG-I, MDA5, and MAVS. Well known agonists of RIG-I should also be tested to determine if this pathway activates GSDME pyroptosis in JEG-3 cells, or if there is something specific to the ZIKV genome that activates a 'non-canonical' RIG-I pathway.

To shore up our conclusions, experiments in Figure 4 were repeated by using RIG-I, MDA5, or MAVS KO JEG-3 cells, according to the reviewer’s suggestion. It was found that the deletion of RIG-I significantly inhibited the activation of GSDME, as well as its upstream caspase3 and caspase8, while the deletion of MDA5 or MAVS only slightly diminished the ZIKV-induced activation of GSDME (Figure 4G and 4H). These findings still support that RIG-I plays an important role in ZIKV-induced GSDME-dependent pyroptosis. Meanwhile, we found that blocking IFN-I signaling did not affect ZIKN 5’ UTR-induced GSDME activation (Figure 4—figure supplement 2D and 2E), suggesting that IFNs signaling is not essential for ZIKV-induced GSDME-dependent pyroptosis.

In addition, RIG-I agonist poly(I:C) and human RIG-IN (active mutant of RIG-I) were used to explore whether the activation of RIG-I can induce GSDME-dependent pyroptosis in the absence of ZIKV infection. As shown in Figure 4—figure supplement 2B and 2C, both poly(I:C) and RIG-IN activated the cleavage of GSDME, indicating that activation of RIG-I is sufficient to induce GSDME-dependent pyroptosis. Taken together, these results demonstrated that RIG-I is the ZIKV sensor upstream of GSDME pyroptosis.

– A surprising omission is an experiment to address the effect of GSDME pyroptosis on viral replication. Although there is not an overt in vivo phenotype on ZIKV replication, viral titers from ZIKV infected WT and GSDME-/- JEG-3 cells should be addressed. This is important for interpreting whether GSDME pyroptosis is only driving pathogenesis, or if GSDME pyroptosis may also play a role in viral clearance.

Thanks for the suggestion. In fact, we have examined the effect of GSDME activation on ZIKV replication by plaque assay. This data has been included in the revised manuscript (Figure 1—figure supplement 1). As shown in the result, KO or overexpression of GSDME did not affect the replication of ZIKV in JEG-3 cells, indicating that GSDME mediated pyroptosis may not play a role in viral clearance.

– Based on the mating scheme, it is not feasible to discern the maternal vs. fetal contributions of GSDME. This is a key question and a limitation of the study design and current results. This should at a minimum be acknowledged in the Discussion.

Thanks for the nice suggestion. We tried to cross *Gsdme*^+/-^ and *Gsdme*^-/-^ mice and hoped to address the maternal vs. fetal contributions of GSDME by comparing pathology in +/- to -/- littermates. However, the pregnancy and litter rates were too low, even though we spent a lot of time and tried many times, we still could not get enough data to draw conclusions. Therefore, the discussion of related issues and the limitations of our in vivo experiments was added in the revised manuscript (Line 467 to 486).

– The in vivo data significantly increases the impact of the manuscript, and I think is a major factor in considering the manuscript for publication in eLife. Although I strongly prefer not to request additional in vivo experiments, the authors currently provide no data to demonstrate that the pathway in mouse trophoblasts is the same as observed in the human JEG-3 cells. Some evidence demonstrating that murine GSDME is being activated by ZIKV via the same pathway as shown for human trophoblasts should be included. Perhaps the most rigorous demonstration would be to test if Rig-I or Casp3 knockout mice phenocopy Gsdme knockouts. However, I would be satisfied if the same pathway was rigorously defined in a mouse trophoblast cell line or primary cells using knockouts or inhibitors.

To verify whether ZIKV could induce pyroptosis via the same pathway in mouse cells as that in human cells, mouse primary trophoblast cells (MTC) were isolated followed by infection of ZIKV. It was found that ZIKV infection induced the activation of GSDME and the release of LDH in MTC, while inhibition of either caspase3 or caspase8 could markedly suppress this process (Figure 5—figure supplement 1), suggesting the same pathway contributing to ZIKV-induced pyroposis in mouse primary trophoblast cells.

For in vivo assay, given the fact that RIG-I plays a crucial role in the innate immune response against flaviviruses, and thus regulate the viral replication (Guo et al., 2018), RIG-I-deficient is likely to affect the replication of ZIKV in mice, which may contribute to the pathological signs of ZIKV infection, so this model may be not suitable for evaluating the function of GSDME in ZIKV-infected cells. Therefore, to address the mechanism in the in vivo model, the role of TNF-α which is known as the upstream regulator of caspase8 was assessed both in vitro and in vivo. Here, we found that RIG KO significantly reduced the expression of TNF-α induced by ZIKV infection (Figure 4K and 4L); blocking TNF-α signaling by using R-7050 (a TNF-α receptor antagonist) restrained the activation of caspase8 and GSDME in ZIKV-infected JEG-3 cells (Figure 3K and 3L). Further in vivo experiments demonstrated that blocking the TNF-α signaling alleviated placental damage and inhibited adverse fetal outcomes caused by ZIKV infection (Figure 6). These findings suggest that ZIKV mediates the activation of caspase8 by promoting the expression of TNF-α through the RIG-I pathway, which eventually leads to the pyroptosis of placenta.

Reviewer #3 (Recommendations for the authors):Given the results in Figure 4, could RIG-I expression in the cell lines in Figure 2 explain differences in pyroptosis?

Thanks for the constructive suggestion. In the current study, we cannot simply determine the contribution of RIG-I expression in pyroptosis during ZIKV infection, and it is possible that a small amount of RIG-I may also play an important role in pathogen recognition. Therefore, we prefer to believe that the different levels of pyroptosis in Figure 2 are caused by the difference in the abundance of GSDME, as studies have fully demonstrated that the abundance of GSDME determines cell death pattern upon stimulation (Wang et al., 2017).

Weight measurements of fetuses would be preferable to the "affected" scoring system in Figure 5C and 6B (which is not described).

According to the suggestion, we have included data of fetal weight in Supplementary Figures (Figure 5—figure supplement 2C and Figure 6—figure supplement 1) in the revised manuscript.

References

(1) Barbeito-Andres J; Pezzuto P; Higa L M; Dias A A; Vasconcelos J M; Santos T M P; Ferreira J; Ferreira R O; Dutra F F; Rossi A D; Barbosa R V; Amorim C K N; De Souza M P C; Chimelli L; Aguiar R S; Gonzalez P N; Lara F A; Castro M C; Molnar Z; Lopes R T et al.; Congenital Zika syndrome is associated with maternal protein malnutrition, *Sci Adv*, 2020, 6(2): eaaw6284

(2) Cox J; Mota J; Sukupolvi-Petty S; Diamond M S; Rico-Hesse R; Mosquito bite delivery of dengue virus enhances immunogenicity and pathogenesis in humanized mice, *J Virol*, 2012, 86(14): 7637-7649

(3) Guo H Y; Zhang X C; Jia R Y; Toll-Like Receptors and RIG-I-Like Receptors Play Important Roles in Resisting Flavivirus, *J Immunol Res*, 2018, 2018: 6106582

(4) Rogers C; Fernandes-Alnemri T; Mayes L; Alnemri D; Cingolani G; Alnemri E S; Cleavage of DFNA5 by caspase-3 during apoptosis mediates progression to secondary necrotic/pyroptotic cell death, *Nat Commun*, 2017, 8: 14128

(5) Styer L M; Kent K A; Albright R G; Bennett C J; Kramer L D; Bernard K A; Mosquitoes inoculate high doses of West Nile virus as they probe and feed on live hosts, *PLoS Pathog*, 2007, 3(9): 1262-1270

(6) Szaba F M; Tighe M; Kummer L W; Lanzer K G; Ward J M; Lanthier P; Kim I J; Kuki A; Blackman M A; Thomas S J; Lin J S; Zika virus infection in immunocompetent pregnant mice causes fetal damage and placental pathology in the absence of fetal infection, *PLoS Pathog*, 2018, 14(4): e1006994

(7) Wang Y; Gao W; Shi X; Ding J; Liu W; He H; Wang K; Shao F; Chemotherapy drugs induce pyroptosis through caspase-3 cleavage of a gasdermin, *Nature*, 2017, 547(7661): 99-103